# Linear Label Ranking with Bounded Noise

**Dimitris Fotakis**
NTUA
fotakis@cs.ntua.gr

**Alkis Kalavasis**
NTUA
kalavasisalkis@mail.ntua.gr

**Vasilis Kontonis**
UW Madison
kontonis@wisc.edu

**Christos Tzamos**
UW Madison
tzamos@wisc.edu

## Abstract

Label Ranking (LR) is the supervised task of learning a sorting function that maps feature vectors $\boldsymbol{x} \in \mathbb{R}^d$ to rankings $\sigma(\boldsymbol{x}) \in \mathbb{S}_k$ over a finite set of $k$ labels. We focus on the fundamental case of learning linear sorting functions (LSFs) under Gaussian marginals: $\boldsymbol{x}$ is sampled from the $d$-dimensional standard normal and the ground truth ranking $\sigma^\star(\boldsymbol{x})$ is the ordering induced by sorting the coordinates of the vector $\boldsymbol{W}^\star \boldsymbol{x}$, where $\boldsymbol{W}^\star \in \mathbb{R}^{k \times d}$ is unknown. We consider learning LSFs in the presence of bounded noise: assuming that a noiseless example is of the form $(\boldsymbol{x}, \sigma^\star(\boldsymbol{x}))$, we observe $(\boldsymbol{x}, \pi)$, where for any pair of elements $i \neq j$, the probability that the order of $i, j$ is different in $\pi$ than in $\sigma^\star(\boldsymbol{x})$ is *at most* $\eta < 1/2$. We design efficient non-proper and proper learning algorithms that learn hypotheses within normalized Kendall's Tau distance $\epsilon$ from the ground truth with $N = \widetilde{O}(d \log(k)/\epsilon)$ labeled examples and runtime poly$(N, k)$. For the more challenging top-$r$ disagreement loss, we give an efficient proper learning algorithm that achieves $\epsilon$ top-$r$ disagreement with the ground truth with $N = \widetilde{O}(dkr/\epsilon)$ samples and poly$(N)$ runtime.

## 1 Introduction

### 1.1 Background and Motivation

Label Ranking (LR) is the problem of learning a hypothesis that maps features to rankings over a finite set of labels. Given a feature vector $\boldsymbol{x} \in \mathbb{R}^d$, a sorting function $\sigma(\cdot)$ maps it to a ranking of $k$ alternatives, i.e., $\sigma(\boldsymbol{x})$ is an element of the symmetric group with $k$ elements, $\mathbb{S}_k$. Assuming access to a training dataset of features labeled with their corresponding rankings, i.e., pairs of the form $(\boldsymbol{x}, \pi) \in \mathbb{R}^d \times \mathbb{S}_k$, the goal of the learner is to find a sorting function $h(\boldsymbol{x})$ that generalizes well over a fresh sample. LR has received significant attention over the years [DSM03, SS07, HFCB08, CH08, FHMB08] due to the large number of applications. For example, ad targeting [DGR$^+$14] is an LR instance where for each user we want to use their feature vector to predict a ranking over ad categories and present them with the most relevant. The practical significance of LR has lead to the development of many techniques based on probabilistic models and instance-based methods [CH08, CDH10], [GDV12, ZLGQ14], decision trees [CHH09], entropy-based ranking trees [RdSRSK15], bagging [AGM17], and random forests [dSSKC17, ZQ18]. However, almost all of these works come without provable guarantees and/or fail to learn in the presence of noise in the observed rankings.

**Linear Sorting Functions (LSFs).** In this work, we focus on the fundamental concept class of Linear Sorting functions [HPRZ03]. A linear sorting function parameterized by a matrix $\boldsymbol{W} \in \mathbb{R}^{k \times d}$ with $k$ rows $\boldsymbol{W}_1, \ldots, \boldsymbol{W}_k$ takes a feature $\boldsymbol{x} \in \mathbb{R}^d$, maps it to $\boldsymbol{W}\boldsymbol{x} = (\boldsymbol{W}_1 \cdot \boldsymbol{x}, \ldots, \boldsymbol{W}_k \cdot \boldsymbol{x}) \in \mathbb{R}^k$ and

36th Conference on Neural Information Processing Systems (NeurIPS 2022).

then outputs an ordering $(i_1, \ldots, i_k)$ of the $k$ alternatives such that $\boldsymbol{W}_{i_1} \cdot \boldsymbol{x} \geq \boldsymbol{W}_{i_2} \cdot \boldsymbol{x} \geq \ldots \geq \boldsymbol{W}_{i_k} \cdot \boldsymbol{x}$. In other words, a linear sorting function ranks the $k$ alternatives (corresponding to rows of $\boldsymbol{W}$) with respect to how well they correlate with the feature $\boldsymbol{x}$. We denote a linear sorting function with parameter $\boldsymbol{W} \in \mathbb{R}^{k \times d}$ by $\sigma_{\boldsymbol{W}}(\boldsymbol{x}) \triangleq \operatorname{argsort}(\boldsymbol{W}\boldsymbol{x})$ where $\operatorname{argsort} : \mathbb{R}^k \to \mathbb{S}_k$ takes as input a vector $(v_1, \ldots, v_k) \in \mathbb{R}^k$, sorts it in decreasing order to obtain $v_{i_1} \geq v_{i_2} \geq \ldots \geq v_{i_k}$ and returns the ordering $(i_1, \ldots, i_k)$.

**Noisy Ranking Distributions.** Learning LSFs in the noiseless setting can be done efficiently by using linear programming. However, the common assumption both in theoretical and in applied works is that the observed rankings are noisy in the sense that they do not always correspond to the ground-truth ranking. We assume that the probability that the order of two elements $i, j$ in the observed ranking $\pi$ is different than their order in the ground-truth ranking $\sigma^\star$ is at most $\eta < 1/2$.

**Definition 1** (Noisy Ranking Distribution). *Fix $\eta \in [0, 1/2)$. An $\eta$-**noisy ranking distribution** $\mathcal{M}(\sigma^\star)$ with ground-truth ranking $\sigma^\star \in \mathbb{S}_k$ is a probability measure over $\mathbb{S}_k$ that, for any $i, j \in [k]$, with $i \neq j$, satisfies $\mathbf{Pr}_{\pi \sim \mathcal{M}(\sigma^\star)}[i \prec_\pi j \mid i \succ_{\sigma^\star} j] \leq \eta$.* [1]

Note that, when $\eta = 0$, we always observe the ground-truth permutation and, in the case of $\eta = 1/2$, we may observe a uniformly random permutation. We remark that most natural ranking distributions satisfy this bounded noise property, e.g., (i) the Mallows model, which is probably the most fundamental ranking distribution (see, e.g., [BM09, LB11, CPS13, ABSV14, BFFSZ19, FKS21, DOS18, LM18, MW20, LM21] for a small sample of this line of research) and (ii) the Bradley-Terry-Mallows model [Mal57], which corresponds to the ranking distribution analogue of the Bradley-Terry-Luce model [BT52, Luc12] (the most studied pairwise comparisons model; see, e.g., [Hun04, NOS17, APA18] and the references therein). For more details, see Appendix E.

We consider the fundamental setting where the feature vector $\boldsymbol{x} \in \mathbb{R}^d$ is generated by a standard normal distribution and the ground-truth ranking for each sample $\boldsymbol{x}$ is given by the LSF $\sigma_{\boldsymbol{W}^\star}(\boldsymbol{x})$ for some unknown parameter matrix $\boldsymbol{W}^\star \in \mathbb{R}^{k \times d}$. For a fixed $\boldsymbol{x}$, the ranking that we observe comes from an $\eta$-noisy ranking distribution with ground-truth ranking $\sigma_{\boldsymbol{W}^\star}(\boldsymbol{x})$.

**Definition 2** (Noisy Linear Label Ranking Distribution). *Fix $\eta \in [0, 1/2)$ and some ground-truth parameter matrix $\boldsymbol{W}^\star \in \mathbb{R}^{k \times d}$. We assume that the $\eta$-**noisy linear label ranking distribution** $\mathcal{D}$ over $\mathbb{R}^d \times \mathbb{S}_k$ satisfies the following:*

1. *The $\boldsymbol{x}$-marginal of $\mathcal{D}$ is the $d$-dimensional standard normal distribution.*

2. *For any $(\boldsymbol{x}, \pi) \sim \mathcal{D}$, the distribution of $\pi$ conditional on $\boldsymbol{x}$ is an $\eta$-noisy ranking distribution with ground-truth ranking $\sigma_{\boldsymbol{W}^\star}(\boldsymbol{x})$.*

At first sight, the assumption that the underlying $\boldsymbol{x}$-marginal is the standard normal may look too strong. However, for $k = 2$, Definition 2 captures the problem of learning linear threshold functions with Massart noise. Without assumptions for the $\boldsymbol{x}$-marginal, it is known [DGT19, CKMY20, DK20, NT22] that optimal learning of halfspaces under Massart noise requires super-polynomial time (in the Statistical Query model of [Kea98]). On the other hand, a lot of recent works [BZ17, MV19, DKTZ20, ZSA20, ZL21] have obtained efficient algorithms for learning Massart halfspaces under Gaussian marginals. The goal of this work is to provide efficient algorithms for the more general problem of learning LSFs with bounded noise under Gaussian marginals.

## 1.2 Our Results

The main contributions of this paper are the first efficient algorithms for learning LSFs with bounded noise with respect to Kendall's Tau distance and top-$r$ disagreement loss.

**Learning in Kendall's Tau Distance.** The most standard metric in rankings [SSBD14] is Kendall's Tau (KT) distance which, for two rankings $\pi, \tau \in \mathbb{S}_k$, measures the fraction of pairs $(i, j)$ on which they disagree. That is, $\Delta_{\mathrm{KT}}(\pi, \tau) = \sum_{i \prec_\pi j} \mathbf{1}\{i \succ_\tau j\} / \binom{k}{2}$. Our first result is an efficient learning algorithm that, given samples from an $\eta$-noisy linear label ranking distribution $\mathcal{D}$, computes

---

[1] We use $i \succ_\pi j$ (resp. $i \prec_\pi j$) to denote that the element $i$ is ranked higher (resp. lower) than $j$ according to the ranking $\pi$.

a parameter matrix $\boldsymbol{W}$ that ranks the alternatives almost optimally with respect to the KT distance from the ground-truth ranking $\sigma_{\boldsymbol{W}^\star}(\cdot)$.

**Theorem 1** (Learning LSFs in KT Distance). *Fix $\eta \in [0, 1/2)$ and $\epsilon, \delta \in (0, 1)$. Let $\mathcal{D}$ be an $\eta$-noisy linear label ranking distribution satisfying the assumptions of Definition 2 with ground-truth LSF $\sigma_{\boldsymbol{W}^\star}(\cdot)$. There exists an algorithm that draws $N = \widetilde{O}\left(\frac{d}{\epsilon(1-2\eta)^6}\log(k/\delta)\right)$ samples from $\mathcal{D}$, runs in sample-polynomial time, and computes a matrix $\boldsymbol{W} \in \mathbb{R}^{k \times d}$ such that, with probability at least $1 - \delta$,*

$$\mathop{\mathbf{E}}_{\boldsymbol{x} \sim \mathcal{N}_d}[\Delta_{\mathrm{KT}}(\sigma_{\boldsymbol{W}}(\boldsymbol{x}), \sigma_{\boldsymbol{W}^\star}(\boldsymbol{x}))] \leq \epsilon\,.$$

Theorem 1 gives the first efficient algorithm with provable guarantees for the supervised problem of learning noisy linear rankings. We remark that the sample complexity of our learning algorithm is qualitatively optimal (up to logarithmic factors) since, for $k = 2$, our problem subsumes learning a linear classifier with Massart noise [2] for which $\Omega(d/\epsilon)$ are known to be information theoretically necessary [MN06]. Moreover, our learning algorithm is *proper* in the sense that it computes a linear sorting function $\sigma_{\boldsymbol{W}}(\cdot)$. As opposed to improper learners (see also Section 1.3), a proper learning algorithm gives us a compact representation (storing $\boldsymbol{W}$ requires $O(kd)$ memory) of the sorting function that allows us to efficiently compute (with runtime $O(kd + k \log k)$) the ranking corresponding to a fresh datapoint $\boldsymbol{x} \in \mathbb{R}^d$.

**Learning in top-$r$ Disagreement.** We next present our learning algorithm for the top-$r$ metric formally defined as $\Delta_{\mathrm{top}-r}(\pi, \tau) = \mathbf{1}\{\pi_{1..r} \neq \tau_{1..r}\}$, where by $\pi_{1..r}$ we denote the ordering on the first $r$ elements of the permutation $\pi$. The top-$r$ metric is a disagreement metric in the sense that it takes binary values and for $r = 1$ captures the standard (multiclass) top-1 classification loss. We remark that, in contrast with the top-$r$ classification loss, which only requires the predicted label to be in the top-$r$ predictions of the model, the top-$r$ ranking metric that we consider here requires that the model puts *the same elements in the same order* as the ground truth in the top-$r$ positions. The top-$r$ ranking is well-motivated as, for example, in ad targeting (discussed in Section 1.1) we want to be accurate on the top-$r$ ad categories for a user so that we can diversify the content that they receive.

**Theorem 2** (Learning LSFs in top-$r$ Disagreement). *Fix $\eta \in [0, 1/2)$, $r \in [k]$ and $\epsilon, \delta \in (0, 1)$. Let $\mathcal{D}$ be an $\eta$-noisy linear label ranking distribution satisfying the assumptions of Definition 2 with ground-truth LSF $\sigma_{\boldsymbol{W}^\star}(\cdot)$. There exists an algorithm that draws $N = \widetilde{O}\left(\frac{drk}{\epsilon(1-2\eta)^6}\log(1/\delta)\right)$ samples from $\mathcal{D}$, runs in sample-polynomial time and computes a matrix $\boldsymbol{W} \in \mathbb{R}^{k \times d}$ such that, with probability at least $1 - \delta$,*

$$\mathop{\mathbf{E}}_{\boldsymbol{x} \sim \mathcal{N}_d}[\Delta_{\mathrm{top}-r}(\sigma_{\boldsymbol{W}}(\boldsymbol{x}), \sigma_{\boldsymbol{W}^\star}(\boldsymbol{x}))] \leq \epsilon\,.$$

As a direct corollary of our result, we obtain a proper algorithm for learning the top-1 element with respect to the standard 0-1 loss that uses $\widetilde{O}(kd)$ samples. In fact, for small values of $r$, i.e., $r = O(1)$, our sample complexity is essentially tight. It is known that $\Theta(kd)$ samples are information theoretically necessary [Nat89] for top-1 classification. [3] For the case $r = k$, i.e., when we want to learn the whole ranking with respect to the 0-1 loss, our sample complexity is $O(k^2 d)$. However, using arguments similar to [DSBDSS11], one can show that in fact $O(dk)$ ranking samples are sufficient in order to learn the whole ranking with respect to the 0-1 loss. In this case, it is unclear whether a better sample complexity can be achieved with an efficient algorithm and we leave this as an interesting open question for future work.

## 1.3 Our Techniques

**Learning in Kendall's Tau distance.** Our proper learning algorithm consists of two steps: an improper learning algorithm that decomposes the ranking problem to $O(k^2)$ binary linear classification problems and a convex (second order conic) program that "compresses" the $k^2$ linear classifiers

---

[2]Notice that in this case Kendall's Tau distance is simply the standard 0-1 binary loss.

[3]Strictly speaking, those lower bounds do not directly apply in our setting because our labels are whole rankings instead of just the top classes but, in the Appendix D, we show that we can adapt the lower bound technique of [DSBDSS11] to obtain the same sample complexity lower bound for our ranking setting.

to obtain a $k \times d$ matrix $\boldsymbol{W}$. Our improper learning algorithm splits the ranking learning problem into $O(k^2)$ binary, $d$-dimensional linear classification problems with Massart noise. In particular, for every pair of elements $i, j \in [k]$, each binary classification task asks whether element $i$ is ranked higher than element $j$ in the ground-truth permutation $\sigma_{\boldsymbol{W}^\star}(\boldsymbol{x})$. As we already discussed, we have that, under the Gaussian distribution, there exist efficient Massart learning algorithms [BZ17, MV19, DKTZ20, ZSA20, ZL21] that can recover linear classifiers $\mathrm{sgn}(\boldsymbol{v}_{ij} \cdot \boldsymbol{x})$ that correctly order the pair $i, j$ for all $\boldsymbol{x}$ apart from a region of $O(\epsilon)$-Gaussian mass. However, we still need to aggregate the results of the *approximate* binary classifiers in order to obtain a ranking of the $k$ alternatives for each $\boldsymbol{x}$. We first show that we can design a "voting scheme" that combines the results of the binary classifiers using an efficient constant factor approximation algorithm for the Minimum Feedback Arc Set (MFAS) problem [ACN08]. This gives us an efficient but improper algorithm for learning LSFs in Kendall's Tau distance. In order to obtain a proper learning algorithm, we further "compress" the $O(k^2)$ approximate linear classifiers with normal vectors $\boldsymbol{v}_{ij}$ and obtain a matrix $\boldsymbol{W} \in \mathbb{R}^{k \times d}$ with the property that the difference of every two rows $\boldsymbol{W}_i - \boldsymbol{W}_j$ is $O(\epsilon)$-close to the vector $\boldsymbol{v}_{ij}$. More precisely, we show that, given the linear classifiers $\boldsymbol{v}_{ij} \in \mathbb{R}^d$, we can efficiently compute a matrix $\boldsymbol{W} \in \mathbb{R}^{k \times d}$ such that the following angle distance with $\boldsymbol{W}^\star$ is small:

$$d_{\mathrm{angle}}(\boldsymbol{W}, \boldsymbol{W}^\star) \triangleq \max_{i,j} \theta(\boldsymbol{W}_i - \boldsymbol{W}_j, \boldsymbol{W}_i^\star - \boldsymbol{W}_j^\star) \leq O(\epsilon) \,. \tag{1}$$

It is not hard to show that, as long as the above angle metric is at most $O(\epsilon)$, then (in expectation over the standard Gaussian) Kendall's Tau distance between the LSFs is also $O(\epsilon)$. A key technical difficulty that we face in this reduction is bounding the "condition number" of the convex (second order conic) program that finds the matrix $\boldsymbol{W}$ given the vectors $\boldsymbol{v}_{ij}$, see Claim 2. Finally, we remark that the proper learning algorithm of Theorem 1 results in a compact and efficient sorting function that requires: (i) storing $O(k)$ weight vectors as opposed to the initial $O(k^2)$ vectors of the improper learner; and (ii) evaluating $k$ inner products with $\boldsymbol{x}$ to find its ranking (instead of $O(k^2)$).

**Learning in top-$r$ Disagreement.** We next turn our attention to the more challenging top-$r$ ranking disagreement metric. In particular, suppose that we are interested in recovering only the top element of the ranking. One approach would be to directly use the improper learning algorithm for this task and ask for KT distance of order roughly $\epsilon/k^2$. The resulting hypothesis would produce good predictions for the top element but the required sample complexity would be $O(dk^2)$. While it seems that training $O(k^2)$ $d$-dimensional binary classifiers inherently requires $O(dk^2)$ samples, we show that, using the proper KT distance learning algorithm of Theorem 1, we can also obtain improved sample complexity results for the top-$r$ metric. Our main technical contribution here is a novel estimate of the top-$r$ disagreement in terms of the angle metric. In general, one can show that the top-$r$ disagreement is at most $O(k^2) \, d_{\mathrm{angle}}(\boldsymbol{W}, \boldsymbol{W}^\star)$. We significantly sharpen this estimate by showing the following lemma.

**Lemma 1** (Top-$r$ Disagreement via Parameter Distance). *Consider two matrices* $\boldsymbol{W}, \boldsymbol{W}^\star \in \mathbb{R}^{k \times d}$ *and let* $\mathcal{N}_d$ *be the standard Gaussian in $d$ dimensions. We have that*

$$\Pr_{\boldsymbol{x} \sim \mathcal{N}_d}[\sigma_{1..r}(\boldsymbol{W}\boldsymbol{x}) \neq \sigma_{1..r}(\boldsymbol{W}^\star\boldsymbol{x})] \leq \widetilde{O}(kr) \, d_{\mathrm{angle}}(\boldsymbol{W}, \boldsymbol{W}^\star) \,.$$

We remark that Lemma 1 is a general geometric tool that we believe will be useful in other distribution-specific multiclass learning settings. The proof of Lemma 1 mainly relies on geometric Gaussian surface area computations that we believe are of independent interest. For the details, we refer the reader to Section 4. An interesting question with a convex-geometric flavor is whether the sharp bound of Lemma 1 also holds under the more general class of isotropic log-concave distributions.

## 1.4  Related Work

**Robust Supervised Learning.** We start with a summary of prior work on PAC learning with Massart noise. The Massart noise model was formally defined in [MN06] but similar variants had been defined by Vapnik, Sloan and Rivest [Vap06, Slo88, Slo92, RS94, Slo96]. This model is a strict extension of the Random Classification Noise (RCN) model [AL88], where the label noise is uniform, i.e., context-independent and is a special case of the agnostic model [Hau18, KSS94], where the label noise is fully adversarial and computational barriers are known to exist [GR09, FGKP06, Dan16, DKZ20, GGK20, DKPZ21, HSSVG22]. Our work partially builds upon on the algorithmic task of

PAC learning halfspaces with Massart noise [BH20]. In the distribution-independent setting, known efficient algorithms [DGT19, CKMY20, DKT21] achieve error $\eta + \epsilon$ and the works of [DK20, NT22] indicate that this error bound is the best possible in the Statistical Query model [Kea98]. This lower bound motivates the study of the distribution-specific setting (which is also the case of our work). There is an extensive line of work in this direction: [ABHU15, ABHZ16, YZ17, ZLC17, BZ17, MV19, DKTZ20, ZSA20, ZL21] with the currently best algorithms succeeding for all $\eta < 1/2$ with a sample and computational complexity $\mathrm{poly}(d, 1/\epsilon, 1/(1 - 2\eta))$ under a class of distributions including isotropic log-concave distributions. For details, see [DKK+21]. In this work we focus on Gaussian marginals but some of our results extend to larger distribution classes.

**Label Ranking.** Our work lies in the area of Label Ranking, which has received significant attention over the years [SS07, HFCB08, CH08, HPRZ03, FHMB08, DSM03]. There are multiple approaches for tackling this problem (see [VG10], [ZLY+14]). Some of them are based on probabilistic models [CH08, CDH10, GDV12, ZLGQ14] or may be tree based, such as decision trees [CHH09], entropy based ranking trees and forests [RdSRSK15, dSSKC17], bagging techniques [AGM17] and random forests [ZQ18]. There are also works focusing on supervised clustering [GDGV13]. Finally, [CH08, CDH10, CHH09] adopt an instance-based approaches using nearest neighbors approaches. The above results are industrial. From a theoretical perspective, LR has been mainly studied from a statistical learning theory framework [CV20, CKS18, KGB18, KCS17]. [FKP21] provide some computational guarantees for the performance of decision trees in the noiseless case and some experimental results on the robustness of random forests to noise. The setting of [DGR+14] is close to ours but is investigated from an experimental standpoint. We remark that while reducing LR to multiple binary classification tasks has been used in prior literature [HFCB08, CH12, FKP21], standard reductions can not tolerate noise in rankings (nevertheless, from an experimental perspective, e.g., random forests seem robust to noise but lack formal theoretical guarantees). Our reduction crucially relies on the existence of efficient learning algorithms for binary linear classification with Massart noise.

## 2 Notation and Preliminaries

**General Notation.** We use $\widetilde{O}(\cdot)$ to omit poly-logarithmic factors. A learning algorithm has sample-polynomial runtime if it runs in time polynomial in the size of the description of the input training set. We denote vectors by boldface $\boldsymbol{x}$ (with elements $x_i$) and matrices with $\boldsymbol{W}$, where we let $\boldsymbol{W}_i \in \mathbb{R}^d$ denote the $i$-th row of $\boldsymbol{W} \in \mathbb{R}^{k \times d}$ and $W_{ij}$ its elements. We denote $\boldsymbol{a} \cdot \boldsymbol{b}$ the inner product of two vectors and $\theta(\boldsymbol{a}, \boldsymbol{b})$ their angle. Let $\mathcal{N}_d$ denote the $d$-dimensional standard normal and $\Gamma(\cdot)$ the Gaussian surface area.

**Rankings.** We let $\mathrm{argsort}_{i \in [k]} \boldsymbol{v}$ denote the ranking of $[k]$ in decreasing order according to the values of $\boldsymbol{v}$. For a ranking $\pi$, we let $\pi(i)$ denote the position of the $i$-th element. If $\pi = \pi(\boldsymbol{x})$, we may also write $\pi(\boldsymbol{x})(i)$ to denote the position of $i$. We often refer to the elements of a ranking as *alternatives*. For a ranking $\sigma$, we let $\sigma_{1..r}$ denote the top-$r$ part of $\sigma$. When $\sigma = \sigma(\boldsymbol{x})$, we may also write $\sigma_{1..r}(\boldsymbol{x})$ and $\sigma_\ell(\boldsymbol{x})$ will be the alternative at the $\ell$-th position. We let $\Delta_{\mathrm{KT}}$ denote the (normalized) KT distance, i.e., $\Delta_{\mathrm{KT}}(\pi, \tau) = \sum_{i \prec_\pi j} \mathbf{1}\{i \succ_\tau j\}/\binom{k}{2}$ for $\pi, \tau \in \mathbb{S}_k$.

## 3 Learning in KT distance: Theorem 1

In this section, we present the main tools required to obtain our proper learning algorithm of Theorem 1. Our proper algorithm adopts a two-step approach: it first invokes an efficient *improper* algorithm which, instead of a linear sorting function (i.e., a matrix $\boldsymbol{W} \in \mathbb{R}^{k \times d}$), outputs a list of $O(k^2)$ linear classifiers. We then design a novel convex program in order to find the matrix $\boldsymbol{W}$ satisfying the guarantees of Theorem 1. Let us begin with the improper learner for LSFs with bounded noise with respect to the KT distance, whose description can be found in Algorithm 1.

### 3.1 Improper Learning Algorithm

Let us assume that the target function is $\sigma^\star(\boldsymbol{x}) = \sigma_{\boldsymbol{W}^\star}(\boldsymbol{x}) = \mathrm{argsort}(\boldsymbol{W}^\star \boldsymbol{x})$ for some $\boldsymbol{W}^\star \in \mathbb{R}^{k \times d}$.

**Step 1: Binary decomposition and Noise Structure.** For each drawn example $(\boldsymbol{x}, \pi)$ from the $\eta$-noisy linear label ranking distribution $\mathcal{D}$ (see Definition 2), we create $\binom{k}{2}$ binary examples $(\boldsymbol{x}, y_{ij})$

---

**Algorithm 1** Non-proper Learning Algorithm `ImproperLSF`

---

`Input:` Training set $T = \{(\boldsymbol{x}^t, \pi^t)\}_{t \in [N]}, \epsilon, \delta \in (0, 1), \eta \in [0, 1/2]$
`Output:` Sorting function $h : \mathbb{R}^d \to \mathbb{S}_k$

For any $1 \le i < j \le k$, create $T_{ij} = \{(\boldsymbol{x}^t, \text{sgn}(\pi^t(i) - \pi^t(j)))\}$
For any $1 \le i < j \le k$, compute $\boldsymbol{v}_{ij} = \texttt{MassartLTF}(T_{ij}, \frac{\epsilon}{4}, \frac{\delta}{10k^2}, \eta)$      ▷ See Appendix A.1
`Ranking Phase:` Given $\boldsymbol{x} \in \mathbb{R}^d$:
    (a) Construct directed graph $G$ with $V(G) = [k]$ and edges $e_{i \to j}$ only if $\boldsymbol{v}_{ij} \cdot \boldsymbol{x} > 0 \; \forall i \ne j$
    (b) Output $h(\boldsymbol{x}) = \texttt{MFAS}(G)$      ▷ See Appendix A.1

---

with $y_{ij} = \text{sgn}(\pi(i) - \pi(j))$ for any $1 \le i < j \le k$. We have that

$$\Pr_{(\boldsymbol{x}, \pi) \sim \mathcal{D}} \left[ y_{ij} \cdot \text{sgn}((\boldsymbol{W}_i^\star - \boldsymbol{W}_j^\star) \cdot \boldsymbol{x}) < 0 \mid \boldsymbol{x} \right] = \Pr_{\pi \sim \mathcal{M}(\sigma^\star(\boldsymbol{x}))} \left[ \pi(i) < \pi(j) \mid \boldsymbol{W}_i^\star \cdot \boldsymbol{x} < \boldsymbol{W}_j^\star \cdot \boldsymbol{x} \right].$$

Since $\mathcal{M}(\sigma^\star(\boldsymbol{x}))$ is an $\eta$-noisy ranking distribution (see Definition 1), we get that the above quantity is at most $\eta < 1/2$. Therefore, each sample $(\boldsymbol{x}, y_{ij})$ can be viewed as a sample from a distribution $\mathcal{D}_{ij}$ with Gaussian $\boldsymbol{x}$-marginal, optimal linear classifier $\text{sgn}((\boldsymbol{W}_i^\star - \boldsymbol{W}_j^\star) \cdot \boldsymbol{x})$, and Massart noise $\eta$. Hence, we have reduced the task of learning noisy LSFs to a number of $\binom{k}{2}$ sub-problems concerning the learnability of halfspaces in the presence of bounded (Massart) noise.

**Step 2: Solving Binary Sub-problems.** We can now apply the algorithm `MassartLTF` for LTFs with Massart noise under standard Gaussian marginals [ZSA20] (for details, see Appendix A.1): for all the pairs of alternatives $1 \le i < j \le k$ with accuracy parameter $\epsilon'$, confidence $\delta' = O(\delta/k^2)$, and a total number of $N = \widetilde{\Omega} \left( \frac{d}{\epsilon'(1-2\eta)^6} \log(k/\delta) \right)$ i.i.d. samples from $\mathcal{D}$, we can obtain a collection of linear classifiers with normal vectors $\boldsymbol{v}_{ij}$ for any $i < j$. We remark that each one of these halfspaces $\boldsymbol{v}_{ij}$ achieves $\epsilon$ disagreement with the ground-truth halfspaces $\boldsymbol{W}_i^\star - \boldsymbol{W}_j^\star$ with high probability, i.e.,

$$\Pr_{\boldsymbol{x} \sim \mathcal{N}_d} [\text{sgn}(\boldsymbol{v}_{ij} \cdot \boldsymbol{x}) \ne \text{sgn}((\boldsymbol{W}_i^\star - \boldsymbol{W}_j^\star) \cdot \boldsymbol{x})] \le \epsilon'.$$

**Step 3: Ranking Phase.** We now have to aggregate the linear classifiers and compute a single sorting function $h : \mathbb{R}^d \to \mathbb{S}_k$. Given an example $\boldsymbol{x}$, we create the tournament graph $G$ with $k$ nodes that contains a directed edge $e_{i \to j}$ if $\boldsymbol{v}_{ij} \cdot \boldsymbol{x} > 0$. If $G$ is acyclic, we output the induced permutation; otherwise, the graph contains cycles which should be eliminated. In order to output a ranking, we remove cycles from $G$ with an efficient, 3-approximation algorithm for MFAS [ACN08, VZW09]. Hence, the output $h(\boldsymbol{x})$ and the true target $\sigma^\star(\boldsymbol{x})$ will have $\mathbf{E}_{\boldsymbol{x} \sim \mathcal{N}_d}[\Delta_{\text{KT}}(h(\boldsymbol{x}), \sigma^\star(\boldsymbol{x}))] \le \epsilon' + 3\epsilon' = 4\epsilon'$. This last equation indicates why a constant factor approximation algorithm suffices for our purposes – we can always pick $\epsilon' = \epsilon/4$ and complete the proof. For details, see Appendix A.1.

### 3.2 Proper Learning Algorithm: Theorem 1

Having obtained the improper learning algorithm, we can now describe our proper Algorithm 2. Initially, the algorithm starts similarly with the improper learner and obtains a collection of binary linear classifiers. The crucial idea is the next step: the design of an appropriate convex program which will efficiently give the matrix $\boldsymbol{W}$. We proceed with the details. For the proof, see Appendix A.2.

---

**Algorithm 2** Proper Learning Algorithm `ProperLSF`

---

`Input:` Training set $T = \{(\boldsymbol{x}^t, \pi^t)\}_{t \in [N]}, \epsilon, \delta \in (0, 1), \eta \in [0, 1/2]$
`Output:` Linear Sorting function $h : \mathbb{R}^d \to \mathbb{S}_k$, i.e., $h(\cdot) = \sigma_{\boldsymbol{W}}(\cdot)$ for some matrix $\boldsymbol{W} \in \mathbb{R}^{k \times d}$

Compute $(\boldsymbol{v}_{ij})_{1 \le i < j \le k} = \texttt{ImproperLSF}(T, \epsilon, \delta, \eta)$      ▷ See Algorithm 1
Setup the CP 1 and compute $\boldsymbol{W} = \texttt{Ellipsoid(CP)}$      ▷ See Appendix A.2
`Ranking Phase:` Given $\boldsymbol{x} \in \mathbb{R}^d$, output $h(\boldsymbol{x}) = \text{argsort}(\boldsymbol{W}\boldsymbol{x})$

---

**Step 1: Calling Non-proper Learners.** As a first step, the algorithm calls Algorithm 1 with parameters $\epsilon, \delta$ and $\eta \in [0, 1/2]$ and obtains a list of linear classifiers with normal vectors $\boldsymbol{v}_{ij}$ for $i < j$. Without loss of generality, assume that $\|\boldsymbol{v}_{ij}\|_2 = 1$.

**Step 2: Designing and Solving the CP 1.** Our main goal is to find a matrix $\boldsymbol{W}$ whose LSF is close to the true target in KT distance. We show the following lemma that connects the KT distance between two LSFs with the angle metric $d_{\text{angle}}(\cdot, \cdot)$ defined in Eq. (1). The proof can be found in the Appendix A.2.

**Lemma 2.** *For $\boldsymbol{W}, \boldsymbol{W}^\star \in \mathbb{R}^{k \times d}$, it holds $\mathbf{E}_{\boldsymbol{x} \sim \mathcal{N}_d}[\Delta_{\text{KT}}(\sigma_{\boldsymbol{W}}(\boldsymbol{x}), \sigma_{\boldsymbol{W}^\star}(\boldsymbol{x}))] \leq d_{\text{angle}}(\boldsymbol{W}, \boldsymbol{W}^\star)$.*

The above lemma states that, for our purposes, it suffices to control the $d_{\text{angle}}$ metric between the guess $\boldsymbol{W}$ and the true matrix $\boldsymbol{W}^\star$. It turns out that, given the binary classifiers $\boldsymbol{v}_{ij}$, we can design a convex program whose solution will satisfy this property. Thinking of the binary classifier $\boldsymbol{v}_{ij}$ as a proxy for $\boldsymbol{W}_i^\star - \boldsymbol{W}_j^\star$, we want each difference $\boldsymbol{W}_i - \boldsymbol{W}_j$ to have small angle with $\boldsymbol{v}_{ij}$ or equivalently to have large correlation with it, i.e., $(\boldsymbol{W}_i - \boldsymbol{W}_j) \cdot \boldsymbol{v}_{ij} \approx \|\boldsymbol{W}_i - \boldsymbol{W}_j\|_2$. To enforce this condition, we can therefore use the second order conic constraint $(\boldsymbol{W}_i - \boldsymbol{W}_j) \cdot \boldsymbol{v}_{ij} \geq (1 - \phi)\|\boldsymbol{W}_i - \boldsymbol{W}_j\|_2$. We formulate the following convex program 1 with variable the matrix $\boldsymbol{W}$:

$$\text{Find} \quad \boldsymbol{W} \in \mathbb{R}^{k \times d}, \quad \|\boldsymbol{W}\|_F \leq 1,$$
$$\text{such that} \quad (\boldsymbol{W}_i - \boldsymbol{W}_j) \cdot \boldsymbol{v}_{ij} \geq (1 - \phi) \cdot \|\boldsymbol{W}_i - \boldsymbol{W}_j\|_2 \quad \text{for any } 1 \leq i < j \leq k, \tag{1}$$

for some $\phi \in (0, 1)$ to be decided. Intuitively, since any $\boldsymbol{v}_{ij}$ has good correlation with $\boldsymbol{W}_i^\star - \boldsymbol{W}_j^\star$ (by the guarantees of the improper learning algorithm) and the CP 1 requires that its solution $\boldsymbol{W}$ similarly correlates well with $\boldsymbol{v}_{ij}$, we expect that $d_{\text{angle}}(\boldsymbol{W}, \boldsymbol{W}^\star)$ will be small. We show that:

**Claim 1.** *The convex program 1 is feasible and any solution $\boldsymbol{W}$ of 1 satisfies $d_{\text{angle}}(\boldsymbol{W}, \boldsymbol{W}^\star) \leq \epsilon$.*

To see this, note that any solution of CP 1 is a matrix $\boldsymbol{W}$ whose angle metric (see Eq. (1)) with the true matrix is small by an application of the triangle inequality between the angles of $(\boldsymbol{v}_{ij}, \boldsymbol{W}_i - \boldsymbol{W}_j)$ and $(\boldsymbol{v}_{ij}, \boldsymbol{W}_i^\star - \boldsymbol{W}_j^\star)$ for any $i \neq j$. We next have to deal with the feasibility of CP 1. Our goal is to determine the value of $\phi$ that makes the CP 1 feasible. For the pair $1 \leq i < j \leq k$, the guess $\boldsymbol{v}_{ij}$ and the true normal vector $\boldsymbol{W}_i^\star - \boldsymbol{W}_j^\star$ satisfy, with high probability,

$$\Pr_{\boldsymbol{x} \sim \mathcal{D}_x}[\text{sgn}(\boldsymbol{v}_{ij} \cdot \boldsymbol{x}) \neq \text{sgn}((\boldsymbol{W}_i^\star - \boldsymbol{W}_j^\star) \cdot \boldsymbol{x})] \leq \epsilon. \tag{2}$$

Under the Gaussian distribution (which is rotationally symmetric), it is well known that the angle $\theta(\boldsymbol{u}, \boldsymbol{v})$ between two vectors $\boldsymbol{u}, \boldsymbol{v} \in \mathbb{R}^d$ is equal to $\pi \cdot \mathbf{Pr}_{\boldsymbol{x} \sim \mathcal{N}_d}[\text{sgn}(\boldsymbol{u} \cdot \boldsymbol{x}) \neq \text{sgn}(\boldsymbol{v} \cdot \boldsymbol{x})]$. Hence, using Eq. (2), we get that the angle between the guess $\boldsymbol{v}_{ij}$ and the true normal vector $\boldsymbol{W}_i^\star - \boldsymbol{W}_j^\star$ is $\theta(\boldsymbol{W}_i^\star - \boldsymbol{W}_j^\star, \boldsymbol{v}_{ij}) \leq c\epsilon$. For sufficiently small $\epsilon$, this bound implies that the cosine of the above angle is of order $1 - (c\epsilon)^2$ and so the following inequality will hold (since $\boldsymbol{v}_{ij}$ is unit):

$$(\boldsymbol{W}_i^\star - \boldsymbol{W}_j^\star) \cdot \boldsymbol{v}_{ij} \geq (1 - 2(c\epsilon)^2) \cdot \|\boldsymbol{W}_i^\star - \boldsymbol{W}_j^\star\|_2.$$

Hence, by setting $\phi = 2(c\epsilon)^2$, the convex program 1 with variables $\boldsymbol{W} \in \mathbb{R}^{k \times d}$ will be feasible; since $\|\boldsymbol{W}^\star\|_F \leq 1$ comes without loss of generality, $\boldsymbol{W}^\star$ will be a solution with probability $1 - \delta$.

Next, we have to control the volume of the feasible region. This is crucial in order to apply the ellipsoid algorithm (for details, see in Appendix A.2.1) and, hence, solve the convex program. We show the following claim (see Appendix A.2.1 for the proof):

**Claim 2.** *There exists $\rho \geq 2^{-\text{poly}(d, k, 1/\epsilon, \log(1/\delta))}$ so that the feasible set of CP 1 with $\phi = O(\epsilon^2)$ contains a ball (with respect to the Frobenius norm) of radius $\rho$.*

Critically, the runtime of the ellipsoid algorithm is *logarithmic* in $1/\rho$. So, the ellipsoid runs in time polynomial in the parameters of the problem and outputs the desired matrix $\boldsymbol{W}$.

## 4 Learning in top-$r$ Disagreement: Theorem 2

In this section we show that the proper learning algorithm of Section 3.2 learns noisy LSFs in the top-$r$ disagreement metric. We have seen that, with $\widetilde{O}(d \log(k)/\epsilon)$ samples, Algorithm 2 of Section 3.2 computes a matrix $\boldsymbol{W}$ such that $d_{\text{angle}}(\boldsymbol{W}, \boldsymbol{W}^\star) \leq \epsilon$, see Claim 1. Let us be more specific. Lemma 2 relates the expected KT distance with the angle metric of the two matrices (see also Equation (1)). Our Algorithm 2 essentially gives an upper bound on this angle metric. When we shift our objective and our goal is to control the top-$r$ disagreement, we can still apply Algorithm 2 which essentially controls the angle metric. The crucial ingredient that is missing is the relation between the loss we

have to control, i.e., the expected top-$r$ disagreement and the angle metric of Equation 1. This relation is presented right after and essentially says that the expected top-$r$ disagreement is at most $O(kr)$ times this angle metric. Hence, in order to get top-$r$ disagreement of order $\epsilon$, it suffices to apply our Algorithm 2 with $\epsilon' = O(\epsilon/(kr))$.

We continue with our main contribution which is the following lemma that connects the top-$r$ disagreement metric with the geometric distance $d_{\mathrm{angle}}(\cdot, \cdot)$, recall Lemma 1. To keep this sketch simple we shall present a sketch of the proof of Lemma 1 for the special case of top-1 classification, which we restate below. The proof of the top-1 case can be found at the Appendix B. The detailed proof of the general case ($r > 1$) can be found in the Appendix C.

**Lemma 3** (Top-1 Disagreement Loss via $d_{\mathrm{angle}}(\cdot, \cdot)$). *Consider two matrices $\boldsymbol{U}, \boldsymbol{V} \in \mathbb{R}^{k \times d}$ and let $\mathcal{N}_d$ be the standard Gaussian in $d$ dimensions. We have that*

$$\Pr_{\boldsymbol{x} \sim \mathcal{N}_d}[\sigma_1(\boldsymbol{U}\boldsymbol{x}) \neq \sigma_1(\boldsymbol{V}\boldsymbol{x})] \leq O\left(k\sqrt{\log k}\right) \, d_{\mathrm{angle}}(\boldsymbol{U}, \boldsymbol{V}).$$

We observe that

$$\Pr_{\boldsymbol{x} \sim \mathcal{N}_d}[\sigma_1(\boldsymbol{U}\boldsymbol{x}) \neq \sigma_1(\boldsymbol{V}\boldsymbol{x})] = \sum_{i \in [k]} \Pr_{\boldsymbol{x} \sim \mathcal{N}_d}[\sigma_1(\boldsymbol{U}\boldsymbol{x}) = i, \sigma_1(\boldsymbol{V}\boldsymbol{x}) \neq i]. \tag{1}$$

We denote by $\mathcal{C}_{\boldsymbol{U}}^{(i)} \triangleq \mathbf{1}\{\boldsymbol{x} : \sigma_1(\boldsymbol{U}\boldsymbol{x}) = i\} = \prod_{j \neq i} \mathbf{1}\{(\boldsymbol{U}_i - \boldsymbol{U}_j) \cdot \boldsymbol{x} \geq 0\}$, i.e., this is the set where the ranking corresponding to $\boldsymbol{U}$ picks $i$ as the top element. Note that $\mathcal{C}_{\boldsymbol{U}}^{(i)}$ is the indicator of a homogeneous polyhedral cone since it can be written as the intersection of homogeneous halfspaces. Using these cones we can rewrite the top-1 disagreement of Eq. (1) as

$$\Pr_{\boldsymbol{x} \sim \mathcal{N}_d}[\sigma_1(\boldsymbol{U}\boldsymbol{x}) \neq \sigma_1(\boldsymbol{V}\boldsymbol{x})] = \sum_{i \in [k]} \Pr_{\boldsymbol{x} \sim \mathcal{N}_d}[C_{\boldsymbol{U}}^{(i)}(\boldsymbol{x}) = 1, C_{\boldsymbol{V}}^{(i)}(\boldsymbol{x}) = 0]. \tag{2}$$

Hence, our task is to control the mass of the disagreement region of two cones. The next Lemma 4 achieves this task and, combined with Eq. (2) directly gives the conclusion of Lemma 3.

Next we work with two general homogeneous polyhedral cones with set indicators $C_1, C_2$:

**Lemma 4** (Cone Disagreement). *Let $C_1, C_2 : \mathbb{R}^d \mapsto \{0, 1\}$ be homogeneous polyhedral cones defined by the $k$ unit vectors $\boldsymbol{v}_1, \ldots, \boldsymbol{v}_k$ and $\boldsymbol{u}_1, \ldots, \boldsymbol{u}_k$ respectively. For some universal constant $c > 0$, it holds that $\Pr_{\boldsymbol{x} \sim \mathcal{N}_d}[C_1(\boldsymbol{x}) \neq C_2(\boldsymbol{x})] \leq c\sqrt{\log k} \, \max_{i \in [k]} \theta(\boldsymbol{v}_i, \boldsymbol{u}_i).$*

**Roadmap of the Proof of Lemma 4:** Assume that we rotate one face of the polyhedral cone $C_1$ by a very small angle $\theta$ to obtain the perturbed cone $C_2$. At a high-level, we expect the probability of the disagreement region between the new cone $C_2$ and $C_1$ to be roughly (this is an underestimation) equal to the size of the perturbation $\theta$ times the (Gaussian) surface area of the face of the convex cone that we perturbed. The Gaussian Surface Area (GSA) of a convex set $A \subset \mathbb{R}^d$, is defined as $\Gamma(A) \triangleq \int_{\partial A} \phi_d(\boldsymbol{x}) d\mu(\boldsymbol{x})$, where $d\mu(\boldsymbol{x})$ is the standard surface measure in $\mathbb{R}^d$ and $\phi_d(\boldsymbol{x}) = (2\pi)^{-d/2} \cdot \exp(-\|\boldsymbol{x}\|_2^2/2)$. In fact, in Claim 3 below, we show that the probability of the disagreement between $C_1$ and $C_2$ is roughly $O(\theta)\Gamma(F_1)\sqrt{\log(1/\Gamma(F_1) + 1)}$, where $F_1$ is the face of cone $C_1$ that we rotated. Now, when we perturb all the faces by small angles (all perturbations are at most $\theta$), we can show (via a sequence of triangle inequalities) that the total probability of the disagreement region is bounded above by the perturbation size $\theta$ times the sum of the Gaussian surface area of every face (times a logarithmic blow-up factor):

$$\Pr_{\boldsymbol{x} \sim \mathcal{N}_d}[C_1(\boldsymbol{x}) \neq C_2(\boldsymbol{x})] \leq O(\theta) \sum_{i=1}^{k} \Gamma(F_i)\sqrt{\log(1/\Gamma(F_i) + 1)}.$$

Surprisingly, for homogeneous convex cones, the above sum cannot grow very fast with $k$. In fact, we show that it can be at most $O(\sqrt{\log k})$. To prove this, we crucially rely on the following convex geometry result showing that the Gaussian surface area of a homogeneous convex cone is $O(1)$ regardless of the number of its faces $k$.

**Lemma 5** ([Naz03]). *Let $C$ be a homogeneous polyhedral cone with $k$ faces $F_1, \ldots, F_k$. Then $C$ has Gaussian surface area $\Gamma(C) = \sum_{i=1}^{k} \Gamma(F_i) \leq 1$.*

Using an inequality similar to the fact that the maximum entropy of a discrete distribution on $k$ elements is at most $\log k$, and, since, from Lemma 5, it holds that $\sum_{i=1}^{k} \Gamma(F_i) \leq 1$, we can show that $\sum_{i=1}^{k} \Gamma(F_i) \sqrt{\log(1/\Gamma(F_i) + 1)} = O(\sqrt{\log k})$. Therefore, with the above lemma we conclude that, if the maximum angle perturbation that we perform on $C_1$ is $\theta$, then the probability of the disagreement region is $O(\theta)$. We next give the formal proof resulting in the upper bound of $O(\sqrt{\log k}\, \theta)$ for the disagreement.

**Single Face Perturbation Bound: Claim 3:** We will use the following notation for the positive orthant indicator $R(\boldsymbol{z}) = \prod_{i=1}^{k} \mathbf{1}\{\boldsymbol{z}_i \geq 0\}$. Notice that the homogeneous polyhedral cone $C_1$ can be written as $C_1(\boldsymbol{x}) = R(\boldsymbol{V}\boldsymbol{x}) = R(\boldsymbol{v}_1 \cdot \boldsymbol{x}, \ldots, \boldsymbol{v}_k \cdot \boldsymbol{x})$. Claim 3 below shows that the disagreement of two cones that differ on a single normal vector is bounded by above by the Gaussian surface area of a particular face $F_1$ times a logarithmic blow-up factor $\sqrt{\log(1/\Gamma(F_1) + 1)}$.

**Claim 3.** *Let $\boldsymbol{v}_1, \ldots, \boldsymbol{v}_k \in \mathbb{R}^d$ and $\boldsymbol{r} \in \mathbb{R}^d$ with $\theta(\boldsymbol{v}_1, \boldsymbol{r}) \leq \theta$ for some sufficiently small $\theta \in (0, \pi/2)$. Let $F_1$ be the face with $\boldsymbol{v}_1 \cdot \boldsymbol{x} = 0$ of the cone $R(\boldsymbol{V}\boldsymbol{x})$ and $c > 0$ be some universal constant. Then,*

$$\Pr_{\boldsymbol{x} \sim \mathcal{N}_d} [R(\boldsymbol{v}_1 \cdot \boldsymbol{x}, \ldots, \boldsymbol{v}_k \cdot \boldsymbol{x}) \neq R(\boldsymbol{r} \cdot \boldsymbol{x}, \boldsymbol{v}_2 \cdot \boldsymbol{x}, \ldots, \boldsymbol{v}_k \cdot \boldsymbol{x})] \leq c \cdot \theta \cdot \Gamma(F_1) \sqrt{\log\left(\frac{1}{\Gamma(F_1)} + 1\right)}.$$

*Proof Sketch of Claim 3.* Since the constraints $\boldsymbol{v}_2 \cdot \boldsymbol{x} \geq 0, \ldots, \boldsymbol{v}_k \cdot \boldsymbol{x} \geq 0$ are common in the two cones, we have that $R(\boldsymbol{v}_1 \cdot \boldsymbol{x}, \ldots, \boldsymbol{v}_k \cdot \boldsymbol{x}) \neq R(\boldsymbol{r} \cdot \boldsymbol{x}, \boldsymbol{v}_2 \cdot \boldsymbol{x}, \ldots, \boldsymbol{v}_k \cdot \boldsymbol{x})$ only when the first "halfspaces" disagree, i.e., when $(\boldsymbol{v}_1 \cdot \boldsymbol{x})(\boldsymbol{r} \cdot \boldsymbol{x}) < 0$. Thus, we have that the LHS probability of Claim 3 is equal to

$$\mathbb{E}_{\boldsymbol{x} \sim \mathcal{N}_d} [R(\boldsymbol{v}_2 \cdot \boldsymbol{x}, \ldots, \boldsymbol{v}_k \cdot \boldsymbol{x}) \cdot \mathbf{1}\{(\boldsymbol{v}_1 \cdot \boldsymbol{x})(\boldsymbol{r} \cdot \boldsymbol{x}) < 0\}] . \tag{3}$$

This expectation contains two terms: the term $R(\boldsymbol{v}_2 \cdot \boldsymbol{x}, \ldots \boldsymbol{v}_k \cdot \boldsymbol{x})$ that contains the last $k - 1$ common constrains of the two cones and the region where the first two halfspaces disagree, i.e., the set $\{\boldsymbol{x} : (\boldsymbol{v}_1 \cdot \boldsymbol{x})(\boldsymbol{r} \cdot \boldsymbol{x}) < 0\}$. In order to upper bound this integral in terms of the angle $\theta$, we observe that (for $\theta$ sufficiently small) it is not hard to show (see Appendix B) that the disagreement region, which is itself a (non-convex) cone, is a subset of the region $\{\boldsymbol{x} : |\boldsymbol{v}_1 \cdot \boldsymbol{x}| \leq 2\theta |\boldsymbol{q} \cdot \boldsymbol{x}|\}$, where $\boldsymbol{q}$ the normalized projection of $\boldsymbol{r}$ onto the orthogonal complement of $\boldsymbol{v}_1$, i.e., $\boldsymbol{q} = \operatorname{proj}_{\boldsymbol{v}_1^{\perp}} \boldsymbol{r} / \|\operatorname{proj}_{\boldsymbol{v}_1^{\perp}} \boldsymbol{r}\|_2$. Therefore, we have that the integral of Eq. (3) is at most

$$\mathbb{E}_{\boldsymbol{x} \sim \mathcal{N}_d} [R(\boldsymbol{v}_2 \cdot \boldsymbol{x}, \ldots, \boldsymbol{v}_k \cdot \boldsymbol{x}) \, \mathbf{1}\{|\boldsymbol{v}_1 \cdot \boldsymbol{x}| \leq 2\theta |\boldsymbol{q} \cdot \boldsymbol{x}|\}] .$$

This is where the definition of the Gaussian surface area appears. In fact, we have to compute the derivative of the above expression (which is a function of $\theta$) with respect to $\theta$ and evaluate it at $\theta = 0$. The idea behind this computation is that we can upper bound probability mass of the cone disagreement, i.e., the term $\Pr_{\boldsymbol{x} \sim \mathcal{N}_d} [R(\boldsymbol{v}_1 \cdot \boldsymbol{x}, \ldots, \boldsymbol{v}_k \cdot \boldsymbol{x}) \neq R(\boldsymbol{r} \cdot \boldsymbol{x}, \boldsymbol{v}_2 \cdot \boldsymbol{x}, \ldots, \boldsymbol{v}_k \cdot \boldsymbol{x})]$ by its derivative with respect to $\theta$ (evaluated at 0) times $\theta$ by introducing $o(\theta)$ error. Hence, it suffices to upper bound the value of this derivative at 0, which is:

$$2 \mathbb{E}_{\boldsymbol{x} \sim \mathcal{N}_d} [R(\boldsymbol{v}_2 \cdot \boldsymbol{x}, \ldots, \boldsymbol{v}_k \cdot \boldsymbol{x}) \, |\boldsymbol{q} \cdot \boldsymbol{x}| \, \delta(|\boldsymbol{v}_1 \cdot \boldsymbol{x}|)] ,$$

where $\delta$ is the Dirac delta function. Notice that, if we did not have the term $|\boldsymbol{q} \cdot \boldsymbol{x}|$, the above expression would be exactly equal to two times the Gaussian surface area of the face with $\boldsymbol{v}_1 \cdot \boldsymbol{x} = 0$, i.e., it would be equal to $2\Gamma(F_1)$. We now show that this extra term of $|\boldsymbol{q} \cdot \boldsymbol{x}|$ can only increase the above surface integral by at most a logarithmic factor. For some $\xi$ to be decided, we have that

$$\mathbb{E}_{\boldsymbol{x} \sim \mathcal{N}_d} [R(\boldsymbol{v}_2 \cdot \boldsymbol{x}, \ldots, \boldsymbol{v}_k \cdot \boldsymbol{x}) \, |\boldsymbol{q} \cdot \boldsymbol{x}| \, \delta(|\boldsymbol{v}_1 \cdot \boldsymbol{x}|)] = \int_{\boldsymbol{x} \in F_1} \phi_d(\boldsymbol{x}) |\boldsymbol{q} \cdot \boldsymbol{x}| d\mu(\boldsymbol{x})$$

$$\leq \int_{\boldsymbol{x} \in F_1} \phi_d(\boldsymbol{x}) |\boldsymbol{q} \cdot \boldsymbol{x}| \mathbf{1}\{|\boldsymbol{q} \cdot \boldsymbol{x}| \leq \xi\} d\mu(\boldsymbol{x}) + \int_{\boldsymbol{x} \in F_1} \phi_d(\boldsymbol{x}) |\boldsymbol{q} \cdot \boldsymbol{x}| \mathbf{1}\{|\boldsymbol{q} \cdot \boldsymbol{x}| \geq \xi\} d\mu(\boldsymbol{x})$$

$$\leq \xi \int_{\boldsymbol{x} \in F_1} \phi_d(\boldsymbol{x}) d\mu(\boldsymbol{x}) + \int_{\boldsymbol{x} \in F_1} \phi_d(\boldsymbol{x}) |\boldsymbol{q} \cdot \boldsymbol{x}| \mathbf{1}\{|\boldsymbol{q} \cdot \boldsymbol{x}| \geq \xi\} d\mu(\boldsymbol{x}) ,$$

where $d\mu(\boldsymbol{x})$ is the standard surface measure in $\mathbb{R}^d$. The first integral above is exactly equal to the Gaussian surface area of the face $F_1$. To bound from above the second term we can use the next claim showing that not a lot of mass of the face $F_1$ can concentrate on the region where $|\boldsymbol{q} \cdot \boldsymbol{x}|$ is very large. Its proof relies on standard Gaussian concentration arguments, and is provided in Appendix B.

**Claim 4.** *It holds that $\int_{\boldsymbol{x} \in F_1} \phi_d(\boldsymbol{x}) |\boldsymbol{q} \cdot \boldsymbol{x}| \mathbf{1}\{|\boldsymbol{q} \cdot \boldsymbol{x}| \geq \xi\} d\mu(\boldsymbol{x}) \leq O(\exp(-\xi^2/2))$.*

Using the above result, we get that

$$\frac{d}{d\theta}\Big( \underset{\boldsymbol{x} \sim \mathcal{N}_d}{\mathbf{E}} [R(\boldsymbol{v}_2 \cdot \boldsymbol{x}, \ldots, \boldsymbol{v}_k \cdot \boldsymbol{x}) \, \mathbf{1}\{|\boldsymbol{v}_1 \cdot \boldsymbol{x}| \leq 2\theta |\boldsymbol{q} \cdot \boldsymbol{x}|\}] \Big)\Big|_{\theta=0} \leq O(\xi)\,\Gamma(F_1) + O(\exp(-\xi^2/2)).$$

By picking $\xi = \Theta(\sqrt{\log(1 + 1/\Gamma(F_1))})$, the result follows since, up to introducing $o(\theta)$ error, we can bound the term $\mathbf{Pr}_{\boldsymbol{x} \sim \mathcal{N}_d}[R(\boldsymbol{v}_1 \cdot \boldsymbol{x}, \ldots, \boldsymbol{v}_k \cdot \boldsymbol{x}) \neq R(\boldsymbol{r} \cdot \boldsymbol{x}, \boldsymbol{v}_2 \cdot \boldsymbol{x}, \ldots, \boldsymbol{v}_k \cdot \boldsymbol{x})]$ by its derivative with respect to $\theta$, evaluated at 0, times $\theta$. $\qquad\square$

**Conclusion.** Our work presents the first theoretical guarantees for (linear) LR with noise and settles interesting directions for future work, as mentioned in Section 1. This paper is theoretical and does not have any negative social impact.

## Acknowledgments and Disclosure of Funding

Dimitris Fotakis and Alkis Kalavasis were supported by the Hellenic Foundation for Research and Innovation (H.F.R.I.) under the "First Call for H.F.R.I. Research Projects to support Faculty members and Researchers and the procurement of high-cost research equipment grant", project BALSAM, HFRIFM17-1424.

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
