# A Learning LSFs with Bounded Noise in Kendall's Tau distance

## A.1 Improperly Learning LSFs with Bounded Noise

We provide an improper learner for LSFs in the presence of bounded noise. We first restate the main result of this section, whose proof relies on a connection between noisy linear label ranking distributions and the Massart noise model.

**Theorem 3** (Non-Proper Learning Algorithm). *Fix $\eta \in [0, 1/2)$ and $\epsilon, \delta \in (0, 1)$. Let $\mathcal{D}$ be an $\eta$-noisy linear label ranking distribution satisfying the assumptions of Definition 2. `ImproperLSF` (Algorithm 1) draws $N = \widetilde{O}\left(\frac{d}{\epsilon(1-2\eta)^6} \log(k/\delta)\right)$ samples from $\mathcal{D}$, runs in $\mathrm{poly}(d, k, 1/\epsilon, \log(1/\delta))$ time and, with probability at least $1 - \delta$, outputs a hypothesis $h : \mathbb{R}^d \to \mathbb{S}_k$ that is $\epsilon$-close in KT distance to the target.*

*Proof.* Assume that the target function is $\sigma^\star(\boldsymbol{x}) = \sigma_{\boldsymbol{W}^\star}(\boldsymbol{x}) = \mathrm{argsort}(\boldsymbol{W}^\star \boldsymbol{x})$ for some unknown matrix $\boldsymbol{W}^\star \in \mathbb{R}^{k \times d}$. Consider a collection of $N$ i.i.d. samples from an $\eta$-noisy linear label ranking distribution $\mathcal{D}$ (see Definition 2) and let $T$ be the associated training set. For each example $(\boldsymbol{x}, \pi) \in T$, we create a list of $\binom{k}{2}$ binary examples $(\boldsymbol{x}, y_{ij})$ with $y_{ij} = \mathrm{sgn}(\pi(i) - \pi(j))$ for any $1 \le i < j \le k$, where $\pi(i)$ denotes the position of the element $i$. Hence, we create the datasets $T_{ij}$ consisting of the binary labeled examples $(\boldsymbol{x}, y_{ij})$. We have that

$$\Pr_{(\boldsymbol{x}, \pi) \sim \mathcal{D}} \left[ y_{ij} \cdot \mathrm{sgn}((\boldsymbol{W}_i^\star - \boldsymbol{W}_j^\star) \cdot \boldsymbol{x}) < 0 \mid \boldsymbol{x} \right] = \Pr_{\pi \sim \mathcal{M}(\sigma^\star(\boldsymbol{x}))} \left[ \pi(i) < \pi(j) \mid \boldsymbol{W}_i^\star \cdot \boldsymbol{x} < \boldsymbol{W}_j^\star \cdot \boldsymbol{x} \right] .$$

Since $\mathcal{M}(\sigma^\star(\boldsymbol{x}))$ is an $\eta$-bounded noise ranking distribution (see Definition 1), we get that

$$\Pr_{\pi \sim \mathcal{M}(\sigma^\star(\boldsymbol{x}))} \left[ \pi(i) < \pi(j) \mid \sigma^\star(\boldsymbol{x})(i) > \sigma^\star(\boldsymbol{x})(j) \right] \le \eta < 1/2 \,,$$

where $\sigma^\star(\boldsymbol{x})(i)$ denotes the position of the element $i$ in the ranking $\sigma^\star(\boldsymbol{x})$. Focusing on the training set $T_{ij}$, we have that the sign $y_{ij}$ is flipped with probability at most $\eta$. So, we have reduced the problem to $\binom{k}{2}$ sub-problems concerning the learnability of halfspaces in the presence of Massart noise. The Massart noise model is a special case of Definition 2 where $k = 2$. Note also that for each training set $T_{ij}$, the features $\boldsymbol{x}$ have the same distribution. We can now apply the following result for LTFs with Massart noise for the standard Gaussian distribution. Recall that the concept class of homogeneous halfspaces (or linear threshold functions) is $\mathcal{C}_{\mathrm{LTF}} = \{h_{\boldsymbol{w}}(\boldsymbol{x}) = \mathrm{sgn}(\boldsymbol{w} \cdot \boldsymbol{x}) : \boldsymbol{w} \in \mathbb{R}^d\}$.

**Lemma 6** (Learning Halfspaces with Massart noise [ZSA20]). *Fix $\eta \in [0, 1/2)$ and let $\epsilon, \delta \in (0, 1)$. Let $\mathcal{D}$ be an $\eta$-noisy linear label ranking distribution satisfying the assumptions of Definition 2 with $k = 2$ (where $\mathcal{C}_{\mathrm{LSF}} = \mathcal{C}_{\mathrm{LTF}}$). There is a computationally efficient algorithm `MassartLTF` that draws $m = O(\frac{d \, \mathrm{polylog}(d)}{\epsilon(1-2\eta)^6} \cdot \log(1/\delta))$ samples from $\mathcal{D}$, runs in $\mathrm{poly}(m)$ time and outputs a linear threshold function $h$ that is $\epsilon$-close to the target linear threshold function $h^\star$ with probability at least $1 - \delta$, i.e., it holds $\Pr_{\boldsymbol{x} \sim \mathcal{N}_d}[h(\boldsymbol{x}) \ne h^\star(\boldsymbol{x})] \le \epsilon$.*

We can invoke the algorithm of Lemma 6 for any alternatives $1 \le i < j \le k$ with accuracy $\epsilon' = O(\epsilon)$, $\delta' = O(\delta/k^2)$ and error rate $\eta < 1/2$[4]. We remark that Lemma 6 returns a halfspace. Each one of the $\binom{k}{2}$ calls will provide a vector $\boldsymbol{v}_{ij} \in \mathbb{R}^d$ such that, with probability at least $1 - \delta'$, it satisfies

$$\Pr_{\boldsymbol{x} \sim \mathcal{N}_d} \left[ \mathrm{sgn}(\boldsymbol{v}_{ij} \cdot \boldsymbol{x}) \ne \mathrm{sgn}((\boldsymbol{W}_i^\star - \boldsymbol{W}_j^\star) \cdot \boldsymbol{x}) \right] \le \epsilon' \,,$$

where the true target halfspace has normal vector $\boldsymbol{W}_i^\star - \boldsymbol{W}_j^\star$. Moreover, for any $i < j$, the algorithm requires that the training set $T_{ij}$ is of size

$$|T_{ij}| = \Omega \left( \frac{d}{\epsilon'} \cdot \frac{1}{(1 - 2\eta)^6} \cdot \log(1/\delta') \right) \,,$$

and, so, a total number of

$$N = \Omega \left( \frac{d}{\epsilon} \cdot \frac{1}{(1 - 2\eta)^6} \cdot \log(k/\delta) \right) \,,$$

---

[4] We can assume that $\eta$ is known without loss of generality.

samples $(\boldsymbol{x}, \pi)$ is required from the distribution $\mathcal{D}$. Given a collection of linear classifiers with normal vectors $\boldsymbol{v}_{ij}$ for any $i < j$, it remains to aggregate them and compute a sorting function $h : \mathbb{R}^d \to \mathbb{S}_k$. To this end, the estimator $h$, given an example $\boldsymbol{x}$, creates the directed complete graph $G$ with $k$ nodes with directed edge $i \to j$ if $\boldsymbol{v}_{ij} \cdot \boldsymbol{x} > 0$. If all the linear classifiers are correct (which occurs with probability $1 - O(\epsilon k^2)$ over $\mathcal{D}_x$ due to the union bound), the graph $G$ is acyclic (since it will match the true directions induced by $\boldsymbol{W}^\star$) and the estimator $h$ outputs the induced permutation. Observe that the KT distance is

$$\frac{1}{\binom{k}{2}} \cdot \mathop{\mathbf{E}}_{\boldsymbol{x} \sim \mathcal{N}_d} \left[ \sum_{1 \leq i < j \leq k} \mathbf{1}\{\mathrm{sgn}(\boldsymbol{v}_{ij} \cdot \boldsymbol{x}) \neq \mathrm{sgn}((\boldsymbol{W}_i^\star - \boldsymbol{W}_j^\star) \cdot \boldsymbol{x})\} \right] \leq \epsilon' \,.$$

Otherwise, the classifiers are inconsistent and $G$ contains cycles. So, the expected number of mistakes in the graph $G$ is $\epsilon k^2$. The estimator in order to output a ranking uses a deterministic constant approximation algorithm for the minimum Feedback Arc Set [ACN08] in order to remove the cycles. For an overview of this fundamental line of research, we refer to [ACN08, VZW09, KMS06].

**Lemma 7** (3-Approximation Algorithm for mimimum FAS (see [VZW09, ACN08])). *There is a deterministic algorithm* `MFAS` *for the minimum Feedback Arc Set on unweighted tournaments with $k$ vertices that outputs orderings with cost less than $3 \cdot \mathrm{OPT}$. The running time is* $\mathrm{poly}(k)$.

In the above, OPT is the minimum number of flips the algorithm should perform. With input the cyclic directed graph $G$ induced by the estimated linear classifiers, the algorithm of Lemma 7 computes, in $\mathrm{poly}(k)$ time, a 3-approximation of the optimal solution (i.e., instead of correcting $\epsilon_0$ directed edges, the algorithm will provide a directed acyclic graph with $3\epsilon_0$ changed edges). Hence, for the hypothesis $h : \mathbb{R}^d \to \mathbb{S}_k$, where $h(\boldsymbol{x})$ is the output of the minimum FAS approximation algorithm with input $G$ ($G$ depends on the input $\boldsymbol{x}$, the randomness of the samples and the internal randomness of the $\binom{k}{2}$ calls of the Massart linear classifiers), and the target function $\sigma^\star(\boldsymbol{x})$, we have that

$$\mathop{\mathbf{E}}_{\boldsymbol{x} \sim \mathcal{N}_d} [\Delta_{KT}(h(\boldsymbol{x}), \sigma^\star(\boldsymbol{x}))] \leq (\epsilon' + 3\epsilon') = 4\epsilon' \,,$$

which completes the proof, by setting $\epsilon' = \epsilon/4$. $\qquad\square$

**Remark 1.** *Consider the following variant of the above procedure: compute the $O(k^2)$ linear classifiers with accuracy $\epsilon' = \epsilon/k^2$: If the induced directed graph is acyclic, output the ranking; otherwise, output a random permutation. With probability $\epsilon$, the KT distance will be of order $k^2$. Hence, one has to draw in total $O(k^4 d/\epsilon)$ samples to make the expected KT distance roughly $O(\epsilon)$. The algorithm of Theorem 3 improves on this approach.*

## A.2 The Proof of Theorem 1: Properly Learning LSFs with Bounded Noise

We first restate the main result of this section.

**Theorem 4** (Proper Learning Algorithm). *Fix $\eta \in [0, 1/2)$ and $\epsilon, \delta \in (0, 1)$. Let $\mathcal{D}$ be an $\eta$-noisy linear label ranking distribution satisfying the assumptions of Definition 2.* `ProperLSF` *(Algorithm 2) draws $N = \widetilde{O}\left( \frac{d}{\epsilon(1-2\eta)^6} \log(k/\delta) \right)$ samples from $\mathcal{D}$, runs in $\mathrm{poly}(d, k, 1/\epsilon, \log(1/\delta))$ time and, with probability at least $1 - \delta$, outputs a Linear Sorting function $h : \mathbb{R}^d \to \mathbb{S}_k$ that is $\epsilon$-close in KT distance to the target.*

We are now ready to provide the proof of our efficient proper learning algorithm for the class of Linear Sorting functions in the presence of bounded noise with respect to the standard Gaussian probability measure.

*Proof.* As a first step, the algorithm calls the improper learning algorithm `ImproperLSF` (Algorithm 1) with parameters $\epsilon, \delta$ and $\eta < 1/2$ and obtains a list of linear classifiers with normal vectors $\boldsymbol{v}_{ij}$ for $i < j$. The utility of this step implies that, with probability at least $1 - \delta$, each one of the classifiers $\epsilon$-learns the associated true halfspace, i.e., it holds

$$\mathop{\mathbf{Pr}}_{\boldsymbol{x} \sim \mathcal{N}_d} [\mathrm{sgn}(\boldsymbol{v}_{ij} \cdot \boldsymbol{x}) \neq \mathrm{sgn}((\boldsymbol{W}_i^\star - \boldsymbol{W}_j^\star) \cdot \boldsymbol{x})] \leq \epsilon \,,$$

where $W^\star$ is the matrix of the target Linear Sorting function. Without loss of generality, assume that $\|v_{ij}\|_2 = 1$. In order to make the learner proper, it suffices to solve the following convex program on $W$:

Find $\quad W \in \mathbb{R}^{k \times d}$, $\hfill$ (1)

such that $\quad (W_i - W_j) \cdot v_{ij} \geq (1 - \phi) \cdot \|W_i - W_j\|_2 \quad$ for any $1 \leq i < j \leq k$, $\qquad$ (CP) $\quad$ (2)

$\qquad \quad \|W\|_F \leq 1$, $\hfill$ (3)

for some $\phi \in (0, 1)$ to be decided. The main key ideas are summarized in the next claim.

**Claim 5.** *The following properties hold true for $\phi = O(\epsilon^2)$ with probability at least $1 - \delta$.*

1. *The convex program 1 is feasible.*

2. *Any solution of the convex program 1 induces an LSF that is $\epsilon$-close in KT distance to the true target $\sigma_{W^\star}(\cdot)$.*

3. *The feasible set of the convex program 1 contains a ball of radius $r = 2^{-\mathrm{poly}(d,k,1/\epsilon,\log(1/\delta))}$ and is contained in a ball of radius 1. Both balls are with respect to the Frobenius norm.*

4. *The convex program 1 can be solved in time $\mathrm{poly}(d, k, 1/\epsilon, \log(1/\delta))$ using the ellipsoid algorithm.*

**Proof of Item 1.** First, we can choose the error $\phi$ so that this convex program is feasible. Let us set $W = W^\star$, where $W^\star$ is the underlying matrix of the target Linear Sorting function $\sigma^\star$ with $\sigma^\star(x) = \mathrm{argsort}(W^\star x)$. Recall that, by the guarantees of the improper learning algorithm, for the pair $1 \leq i < j \leq k$, it holds

$$\Pr_{x \sim \mathcal{N}_d}[\mathrm{sgn}(v_{ij} \cdot x) \neq \mathrm{sgn}((W_i^\star - W_j^\star) \cdot x)] \leq \epsilon. \qquad (4)$$

Since the standard Gaussian is rotationally symmetric, the angle $\theta(u, v)$ between two vectors $u, v \in \mathbb{R}^d$ is equal to $\pi \cdot \Pr_{x \sim \mathcal{N}_d}[\mathrm{sgn}(u \cdot x) \neq \mathrm{sgn}(v \cdot x)]$. Hence, using this observation and Equation (4), we get that the angle between the guess vector $v_{ij}$ and the true normal vector $W_i^\star - W_j^\star$ is

$$\theta(W_i^\star - W_j^\star, v_{ij}) \leq c \cdot \epsilon,$$

for some constant $c > 0$. For sufficiently small $\epsilon$, this bound implies that the cosine of the above angle is of order $1 - (c\epsilon)^2$ and so the following inequality will hold

$$(W_i^\star - W_j^\star) \cdot v_{ij} \geq (1 - 2(c\epsilon)^2) \cdot \|W_i^\star - W_j^\star\|_2,$$

since $v_{ij}$ is unit. Hence, by setting $\phi = 2(c\epsilon)^2$, the convex program with variables $W \in \mathbb{R}^{k \times d}$ will be feasible; $W^\star$ will be a solution with probability $1 - \delta$, where the randomness is over the output of the algorithm dealing with the Massart linear classifiers. Note that we can assume that $\|W^\star\|_F \leq 1$ without loss of generality, since we can divide each row with the Frobenius norm.

**Proof of Item 2.** Let $\widetilde{W}$ be a solution of the convex program. We will make use of the observation that the angle between two vectors is equal to the disagreement of the associated linear threshold functions with respect to the standard normal times $\pi$. Observe that any solution $\widetilde{W}$ to the convex program will satisfy that

$$(\forall i, j) \quad \theta(v_{ij}, \widetilde{W}_i - \widetilde{W}_j) \leq O(\sqrt{\phi}) = c\epsilon.$$

and

$$(\forall i, j) \quad \theta(W_i^\star - W_j^\star, v_{ij}) \leq \epsilon.$$

This implies that

$$d_{\mathrm{angle}}(W^\star, \widetilde{W}) \leq c' \epsilon$$

**Claim 6.** *For the matrices $W, W^\star \in \mathbb{R}^{k \times d}$, it holds that*

$$\mathbb{E}_{x \sim \mathcal{N}_d}[\Delta_{\mathrm{KT}}(\sigma_W(x), \sigma_{W^\star}(x))] \leq d_{\mathrm{angle}}(W, W^\star).$$

*Proof.* We have that

$$\mathbf{E}_{\boldsymbol{x}\sim\mathcal{N}_d}[\Delta_{\mathrm{KT}}(\sigma_{\boldsymbol{W}}(\boldsymbol{x}),\sigma_{\boldsymbol{W}^\star}(\boldsymbol{x}))] = \frac{1}{\binom{k}{2}}\cdot\mathbf{E}_{\boldsymbol{x}\sim\mathcal{N}_d}\Big[\sum_{1\le i<j\le k}\mathbf{1}\{((\boldsymbol{W}_i-\boldsymbol{W}_j)\cdot\boldsymbol{x})((\boldsymbol{W}_i^\star-\boldsymbol{W}_j^\star)\cdot\boldsymbol{x})<0\}$$

$$= \frac{1}{\binom{k}{2}}\cdot\sum_{1\le i<j\le k}\Pr_{\boldsymbol{x}\sim\mathcal{N}_d}[\mathrm{sgn}(\boldsymbol{W}_i-\boldsymbol{W}_j)\cdot\boldsymbol{x}\ne\mathrm{sgn}((\boldsymbol{W}_i^\star-\boldsymbol{W}_j^\star)\cdot\boldsymbol{x})]$$

$$= \frac{1}{\pi}\max_{i,j}\theta(\boldsymbol{W}_i-\boldsymbol{W}_j,\boldsymbol{W}_i^\star-\boldsymbol{W}_j^\star)$$

$$\le d_{\mathrm{angle}}(\boldsymbol{W},\boldsymbol{W}^\star).$$

$\square$

Using the above claim, we get an expected KT distance bound of order $O(\epsilon)$. This gives the desired result.

**Proof of Item 3.** We will make use of the next lemma.

**Lemma 8.** *Fix $\epsilon,\delta\in(0,1)$. Let $\boldsymbol{W}^\star\in\mathbb{R}^{k\times d}$ be the true parameter matrix. There exists a matrix $\widetilde{\boldsymbol{W}}^\star\in\mathbb{R}^{k\times d}$ such that, with probability at least $1-\delta$:*

- $\Pr_{\boldsymbol{x}\sim\mathcal{N}_d}[\mathrm{sgn}((\boldsymbol{W}_i^\star-\boldsymbol{W}_j^\star)\cdot\boldsymbol{x})\ne\mathrm{sgn}((\widetilde{\boldsymbol{W}}_i^\star-\widetilde{\boldsymbol{W}}_j^\star)\cdot\boldsymbol{x})]\le\epsilon$ *for all $i\ne j$, and,*

- $\|\widetilde{\boldsymbol{W}}_i^\star-\widetilde{\boldsymbol{W}}_j^\star\|_2\ge 2^{-\mathrm{poly}(d,k,1/\epsilon,\log(1/\delta))}$ *for any $i\ne j$.*

*Proof of Lemma 8.* The above lemma is a result of the next Appendix A.2.1. In particular, it is a direct implication of Lemma 10 and Corollary 1. $\square$

Note that the above lemma implies that

$$(\forall i,j)\quad\Pr_{\boldsymbol{x}\sim\mathcal{N}_d}[\mathrm{sgn}(\boldsymbol{v}_{ij}\cdot\boldsymbol{x})\ne\mathrm{sgn}((\widetilde{\boldsymbol{W}}_i^\star-\widetilde{\boldsymbol{W}}_j^\star)\cdot\boldsymbol{x})]\le 2\epsilon,$$

with probability at least $1-2\delta$. Hence, up to constants, the analysis concerning the feasibility of the true matrix $\boldsymbol{W}^\star$ (see Item 1) will still hold for $\widetilde{\boldsymbol{W}}^\star$. From now on we can work with this matrix $\widetilde{\boldsymbol{W}}^\star$ which enjoys the "well-conditionedness" property of the second item of the lemma.

We will use the above lemma in order to prove Item 3 which controls the volume of the feasible region: it states that there exist $0<r<R$ so that the feasible region of the convex program contains a ball of radius $r$ and is contained in a ball of radius $R$ (where the balls are with respect to the Frobenius norm). Moreover, $r=2^{-\mathrm{poly}(d,k,1/\epsilon,\log(1/\delta))}$ and $R=1$.

For the chosen $\phi\in(0,1)$, the feasible set contains matrices $\boldsymbol{W}\in\mathbb{R}^{k\times d}$ that satisfy $\|\boldsymbol{W}-\widetilde{\boldsymbol{W}}^\star\|_F\le 2r$, $r$ to be decided. For any $i\ne j$, we have that the following properties hold:

1. $\|\widetilde{\boldsymbol{W}}_i^\star-\widetilde{\boldsymbol{W}}_j^\star\|_2\ge 2^{-\mathrm{poly}(d,k,1/\epsilon,\log(1/\delta))}$ (well-conditionedness).

2. $(\widetilde{\boldsymbol{W}}_i^\star-\widetilde{\boldsymbol{W}}_j^\star)\cdot\boldsymbol{v}_{ij}\ge(1-\phi)\|\widetilde{\boldsymbol{W}}_i^\star-\widetilde{\boldsymbol{W}}_j^\star\|_2$ (feasibility).

3. $\|\boldsymbol{W}-\widetilde{\boldsymbol{W}}^\star\|_F\le 2r$ which implies that $\|\boldsymbol{W}_i-\widetilde{\boldsymbol{W}}_i^\star\|_2\le 2r$ for any $i\in[k]$ (ball around feasible point).

4. $\|\boldsymbol{v}_{ij}\|_2=1$.

Our goal is to prove that for a matrix in the above ball it holds $(\boldsymbol{W}_i-\boldsymbol{W}_j)\cdot\boldsymbol{v}_{ij}\ge(1-\phi)\|\boldsymbol{W}_i-\boldsymbol{W}_j\|_2$.

We have that

$$(\widetilde{\boldsymbol{W}}_i^\star-\widetilde{\boldsymbol{W}}_j^\star)\cdot\boldsymbol{v}_{ij} = (\widetilde{\boldsymbol{W}}_i^\star-\boldsymbol{W}_i)\cdot\boldsymbol{v}_{ij}+(\boldsymbol{W}_j-\widetilde{\boldsymbol{W}}_j^\star)\cdot\boldsymbol{v}_{ij}+(\boldsymbol{W}_i-\boldsymbol{W}_j)\cdot\boldsymbol{v}_{ij}$$

$$\le\|\widetilde{\boldsymbol{W}}_i^\star-\boldsymbol{W}_i\|_2+\|\boldsymbol{W}_j-\widetilde{\boldsymbol{W}}_j^\star\|_2+(\boldsymbol{W}_i-\boldsymbol{W}_j)\cdot\boldsymbol{v}_{ij}$$

$$\le 4r+(\boldsymbol{W}_i-\boldsymbol{W}_j)\cdot\boldsymbol{v}_{ij}.$$

More to that

$$\|W_i - W_j\|_2 = \|W_i - \widetilde{W}_i^\star + \widetilde{W}_i^\star - \widetilde{W}_j^\star + \widetilde{W}_j^\star - W_j\|_2$$
$$\leq \|W_i - \widetilde{W}_i^\star\|_2 + \|\widetilde{W}_i^\star - \widetilde{W}_j^\star\|_2 + \|\widetilde{W}_j^\star - W_j\|_2$$
$$\leq 4r + \|\widetilde{W}_i^\star - \widetilde{W}_j^\star\|_2,$$

and similarly: $\|W_i - W_j\|_2 \geq \|\widetilde{W}_i^\star - \widetilde{W}_j^\star\|_2 - 4r$.

Combining the above inequalities, we get that

$$(W_i - W_j) \cdot v_{ij} \geq (\widetilde{W}_i^\star - \widetilde{W}_j^\star) \cdot v_{ij} - 4r$$
$$\geq (1 - \phi) \|\widetilde{W}_i^\star - \widetilde{W}_j^\star\|_2 - 4r$$
$$\geq (1 - \phi) (\|W_i - W_j\|_2 - 4r) - 4r$$
$$= (1 - \phi) \|W_i - W_j\|_2 - 8r.$$

We pick $r$ sufficiently small and of order $2^{-\mathrm{poly}(d,k,1/\epsilon,\log(1/\delta))}$ and get that $W$ is a feasible solution of the convex program 1. Moreover, we can select $R = 1$ since $\|\widetilde{W}^\star\|_F = 1$ without loss of generality, since we can normalize the row differences of $\widetilde{W}^\star$ with the norm $\|\widetilde{W}^\star\|_F$.

**Proof of Item 4.** We apply the ellipsoid algorithm in order to solve the convex program 1 and compute a matrix $\widetilde{W} \in \mathbb{R}^{k \times d}$. The algorithm `ProperLSF` outputs the linear sorting function $h(\cdot) = \sigma_{\widetilde{W}}(\cdot)$.

**Lemma 9** (Efficiency of the Ellipsoid Algorithm [Vis21])**.** *Suppose that $P \subseteq \mathbb{R}^d$ is a full-dimensional polytope that is contained in a $d$-dimensional Euclidean ball of radius $R > 0$ and contains a $d$-dimensional Euclidean ball of radius $r > 0$. Then, the ellipsoid method outputs a point $\widetilde{x} \in P$ after $O(d^2 \log(R/r))$ iterations. Moreover, every iteration can be implemented in $O(d^2 + T_{\mathrm{sep}})$ time, where $T_{\mathrm{sep}}$ is the time required to answer a single query by the separation oracle.*

Assume that Item 3 holds true. Then the algorithm can be used with $r = 2^{-\mathrm{poly}(d,k,1/\epsilon,\log(1/\delta))}$ and $R = 1$. Hence, the ellipsoid algorithm will provide in time $\mathrm{poly}(d, k, 1/\epsilon, \log(1/\delta))$ a point $\widetilde{W}$ that lies in the feasible region of the convex program 1[5].

$\square$

**Remark 2.** *We remark that both the improper (Algorithm 1) and the proper (Algorithm 2) learning algorithms hold for the more general case where the $x$-marginal lies in the class of isotropic log-concave distributions [LV07]: A distribution $\mathcal{D}_x$ lies inside the class of isotropic log-concave distributions $\mathcal{F}_{\mathrm{LC}}$ over $\mathbb{R}^d$ if $\mathcal{D}_x$ has a probability density function $f$ over $\mathbb{R}^d$ such that $\log f$ is concave, its mean is zero, and its covariance is identity, i.e., $\mathbf{E}_{x \sim \mathcal{D}_x}[xx^\top] = I$.*

### A.2.1 The proof of Lemma 8

We provide the following result.

**Lemma 10.** *Fix $\epsilon, \delta \in (0, 1)$. Let $W^\star \in \mathbb{R}^{k \times d}$ be the true parameter matrix. There exists a matrix $W \in \mathbb{R}^{k \times d}$ such that, with probability at least $1 - \delta$:*

- $\mathbf{Pr}_{x \sim \mathcal{N}_d}[\mathrm{sgn}((W_i^\star - W_j^\star) \cdot x) \neq \mathrm{sgn}((W_i - W_j) \cdot x)] \leq \epsilon$ *for all $i \neq j$, and,*

- *The bit complexity of $W$ is $\mathrm{poly}(k, d, 1/\epsilon, \log(1/\delta))$*

*Proof.* The matrix $W$ will be the output of a linear program that can be used to learn the LSF $\sigma_{W^\star}(\cdot)$ in the noiseless setting.

---

[5]We remark that the runtime will also depend on the time required to answer a single query by the separation oracle. We assume that this time is polynomial in the parameters of our problem and we opt not to track these details in this work.

Consider the unit sphere $\mathcal{S}^{d-1}$ and a $\delta_0$-cover of the unit sphere with parameter $\delta_0 > 0$ to be decided. For any sample $(\boldsymbol{x}, \pi) \sim \mathcal{D}$ of the 0-noisy linear label ranking distribution, i.e., $\boldsymbol{x} \sim \mathcal{N}_d$ and $\pi = \sigma_{\boldsymbol{W}^\star}(\boldsymbol{x})$, we consider the rounded sample $(\widetilde{\boldsymbol{x}}, \pi)$ where $\widetilde{\boldsymbol{x}}$ is obtained by first projecting $\boldsymbol{x} \in \mathbb{R}^d$ to $\mathcal{S}^{d-1}$ and then by obtaining the closest point of $\widehat{x}$ in the cover. The cover's size is $O(1/\delta_0)^d$.

Let us fix $1 \leq i < j \leq k$ and set $y_{ij} = \text{sgn}(\pi(i) - \pi(j))$. For a training set $\{(\boldsymbol{x}^{(t)}, \pi^{(t)})\}_{t \in [N]}$ of size $N$, we create the following linear system $\mathrm{L}_{ij}$ with variables $\boldsymbol{W} \in \mathbb{R}^{k \times d}$:

$$y_{ij}^{(t)} \, (\boldsymbol{W}_i - \boldsymbol{W}_j) \cdot \widetilde{\boldsymbol{x}}^{(t)} \geq 0 \, , \; t \in [N] \quad (\mathrm{L}_{ij}) \, .$$

Consider the concatenation of the linear systems $\mathrm{L} = \cup_{i<j} \mathrm{L}_{ij}$. The number of equations in the linear system of equations $\mathrm{L}$ is $N \cdot \binom{k}{2}$.

We first have to show that, with high probability, the system $\mathrm{L}$ is feasible, i.e., there exists $\boldsymbol{W}$ that satisfies the system's equations. Note that if we replace $\widetilde{\boldsymbol{x}}^{(t)}$ with the original points $\boldsymbol{x}^{(t)}$, the true matrix $\boldsymbol{W}^\star$ is a solution to the system. We now have to study the rounded linear system.

**Claim 7.** *The (rounded) linear system $\mathrm{L}$ is feasible with high probability.*

*Proof.* In order to show the feasibility of $\mathrm{L}$, we will use the anti-concentration properties of the Gaussian.

**Fact 1** ([DKM05]). *Let $\mathcal{P}$ be the standard normal distribution over $\mathbb{R}^d$. For any fixed unit vector $\boldsymbol{a} \in \mathbb{R}^d$ and any $\gamma \leq 1$,*

$$\gamma/4 \leq \Pr_{\boldsymbol{x} \sim \mathcal{P}}\left[|\boldsymbol{a} \cdot \frac{\boldsymbol{x}}{\|\boldsymbol{x}\|_2}| \leq \frac{\gamma}{\sqrt{d}}\right] \leq \gamma \, .$$

Let us focus on the pair $1 \leq i < j \leq k$. We first observe that scaling all samples to lie on the unit sphere does not affect the feasibility of the system. It suffices to focus on that single halfspace with normal vector $\boldsymbol{v}_{ij} = \boldsymbol{W}_i^\star - \boldsymbol{W}_j^\star \in \mathbb{R}^d$ and consider the probability of the event that the collection of the $N$ rounded points $\{\widetilde{\boldsymbol{x}}^{(t)}\}_t$ with labels $\{y_{ij}^{(t)}\}_t$, that come from $N$ Gaussian vectors $\{\boldsymbol{x}^{(t)}\}_t$ which are linearly separable (with labels $\{y_{ij}^{(t)}\}_t$), becomes non-linearly separable. For this it suffices to control the probability that the rounding procedure flips the label of the data point. Using the union bound, we have that, if the rounding has accuracy $\delta_0$, the described bad event has probability

$$\Pr_{\boldsymbol{x}^{(1)}, \dots, \boldsymbol{x}^{(N)} \sim \mathcal{N}_d}[\exists t \in [N] : \text{sgn}(\boldsymbol{v}_{ij} \cdot \widetilde{\boldsymbol{x}}^{(t)}) \neq \text{sgn}(\boldsymbol{v}_{ij} \cdot \boldsymbol{x}^{(t)})] \leq N \cdot \Pr_{\boldsymbol{x} \sim \mathcal{N}_d}[|\boldsymbol{v}_{ij} \cdot \boldsymbol{x}/\|\boldsymbol{x}\|_2| \leq 2\delta_0] \leq N \cdot O(\delta_0\sqrt{d}) \, ,$$

where we remark that the first event is scale invariant and so we can assume that the normal vector is unit, the first inequality follows from the fact that it suffices to control the mass assigned to a strip of width $2\delta_0$ (due to the discretization) and the second inequality follows from Fact 1. We now have to select the discretization. Let $\delta \in (0, 1)$. By choosing $\delta_0 = O(\frac{\delta}{N\sqrt{d}k^2})$, the bad event for all the pairs $i < j$ occurs with probability at most $\delta$, i.e., with probability at least $1 - \delta$, each one of the $N$ drawn i.i.d. samples does not fall in any one of the $\binom{k}{2}$ "bad" strips. $\square$

We can now consider the case that the system $\mathrm{L}$ is feasible (with the target matrix $\boldsymbol{W}^\star$ being a feasible point) that occurs with probability $1 - \delta$. The class of homogenous halfspaces in $d$ dimensions has VC dimension $d$; therefore, the sample complexity of learning halfspaces using ERM is $O((d + \log(1/\delta))/\epsilon)$. Moreover, in the realizable case, we can implement the ERM using e.g., linear programming and find a solution in $\text{poly}(d, 1/\epsilon, \log(1/\delta))$ time. We next focus on the quality of the solution which will give the desired sample complexity.

**Claim 8.** *Assume that the algorithm draws $N = \widetilde{O}(\frac{d + \log(k/\delta)}{\epsilon})$ i.i.d. samples of the form $(\boldsymbol{x}, \pi)$ with $\boldsymbol{x} \sim \mathcal{N}_d$ and $\pi = \sigma_{\boldsymbol{W}^\star}(\boldsymbol{x})$. For any $i \neq j$ and with probability at least $1 - 2\delta$, the solution $\boldsymbol{W}$ of the linear system $\mathrm{L}$ satisfies*

$$\Pr_{\boldsymbol{x} \sim \mathcal{N}_d}[\text{sgn}((\boldsymbol{W}_i^\star - \boldsymbol{W}_j^\star) \cdot \boldsymbol{x}) \neq \text{sgn}((\boldsymbol{W}_i - \boldsymbol{W}_j) \cdot \boldsymbol{x})] \leq \epsilon \, .$$

*Proof.* Since the matrix $\boldsymbol{W}$ satisfies the sub-system $\mathrm{L}_{ij}$, the result follows using a union bound on the events that (i) the linear system is feasible and (ii) the ERM is a successful PAC learner. $\square$

**Claim 9.** *Consider the solution $\boldsymbol{W}$ of the linear system. Then, $\boldsymbol{W}$ has bounded bit complexity of order* $\mathrm{poly}(d, k, 1/\epsilon, \log(1/\delta))$.

*Proof.* We will make use of the following result that relates the size of the input and the output of a linear program using Cramer's rule.

**Lemma 11** ([Sch98, Pap81]). *Let $\boldsymbol{A} \in \mathbb{Z}^{m \times n}, \boldsymbol{b} \in \mathbb{Z}^m, \boldsymbol{c} \in \mathbb{Z}^n$. Consider a linear program $\min \boldsymbol{c} \cdot \boldsymbol{x}$ subject to $\boldsymbol{A}\boldsymbol{x} \leq \boldsymbol{b}$ and $\boldsymbol{x} \geq \boldsymbol{0}$. Let $U$ be the maximum size of $A_{ij}, b_i, c_j$. The output of the linear program has size $O(m(nU + n\log(n)))$ bits.*

We will apply the above lemma (which holds even by dropping the constraint $\boldsymbol{x} \geq \boldsymbol{0}$) to our setting where $\boldsymbol{A}\boldsymbol{w} \geq 0$ where $\boldsymbol{w} = (\boldsymbol{W}_i)_{i \in [k]} \in \mathbb{Q}^{kd}$, i.e., $\boldsymbol{w}$ is the vectorization of the matrix $\boldsymbol{W}$. Moreover, $\boldsymbol{A}$ is the matrix containing the $N$ (rounded) Gaussian samples $\widetilde{\boldsymbol{x}}^{(t)}$. We have that the matrix $\boldsymbol{A}$ has dimension $N\binom{k}{2} \times kd$ and each entry $A_{ij}$ is an integer and has size at most $U = \mathrm{poly}(d, k)$ (since the samples are rounded on the $\delta_0$-cover of the sphere. Recall that the labels $y_{ij}^{(t)} \in \{-1, +1\}$ and $\widetilde{\boldsymbol{x}}^{(t)}$ lie in the unit sphere. In particular, each row of the matrix $\boldsymbol{A}$ has $2d$ non-zero entries and is associated with a tuple $(i, j, t)$ for $1 \leq i < j \leq k$ and $t \in [N]$. Then, it holds that the output has size at most $O(Nk^2(dU + dk\log(dk)))$ bits. So, we get that the output $\boldsymbol{W}$ can be described using at most $\mathrm{poly}(d, k, 1/\epsilon, U, \log(1/\delta)) = \mathrm{poly}(d, k, 1/\epsilon, \log(1/\delta))$ bits (due to the size of the entries of the matrix $\boldsymbol{A}$). $\qquad\square$

Combining the above claims, we conclude the proof. $\qquad\square$

As a corollary of the bounded bit complexity, we obtain the following key result.

**Corollary 1.** *Let $\epsilon > 0$. Assume that $\boldsymbol{W} \in \mathbb{R}^{k \times d}$ has bit complexity at most $\mathrm{poly}(d, k, 1/\epsilon, \log(1/\delta))$. Then, for any $i, j \in [k]$ with $i \neq j$, it holds that $\|\boldsymbol{W}_i - \boldsymbol{W}_j\|_2 > 2^{-\mathrm{poly}(d,k,1/\epsilon,\log(1/\delta))}$.*

*Proof.* First, we can assume that $\boldsymbol{W}_i \neq \boldsymbol{W}_j$ for any $i \neq j$; in case of equal rows, we obtain a low-dimensional instance. Then, since any vector $\boldsymbol{W}_i$ has bounded bit complexity, we have that the difference of any two such vectors, provided that it is non-zero, has a lower bound in its norm, i.e., $\|\boldsymbol{W}_i - \boldsymbol{W}_j\|_2 > 2^{-\mathrm{poly}(d,k,1/\epsilon,\log(1/\delta))}$ for any $i, j \in [k]$. $\qquad\square$

## B  Learning in Top-1 Disagreement from Label Rankings

Let us set $\sigma_1(\boldsymbol{W}\boldsymbol{x}) = \mathrm{argmax}_{i \in [k]} \boldsymbol{W}_i \cdot \boldsymbol{x}$ for $\boldsymbol{x} \in \mathbb{R}^d$. The main result of this section follows.

**Theorem 5** (Proper Top-1 Learning Algorithm). *Fix $\eta \in [0, 1/2)$ and $\epsilon, \delta \in (0, 1)$. Let $\mathcal{D}$ be an $\eta$-noisy linear label ranking distribution satisfying the assumptions of Definition 2. There exists an algorithm that draws $N = O\left(\frac{dk\sqrt{\log k}}{\epsilon(1-2\eta)^6} \log(k/\delta)\right)$ samples from $\mathcal{D}$, runs in $\mathrm{poly}(N)$ time and, with probability at least $1 - \delta$, outputs a Linear Sorting function $h : \mathbb{R}^d \to \mathbb{S}_k$ that is $\epsilon$-close in top-1 disagreement to the target.*

*Proof.* Note that the `MassartLTF` algorithm (see Lemma 6) has the guarantee that it returns a vector $\boldsymbol{w}$ so that
$$\Pr_{\boldsymbol{x} \sim \mathcal{N}_d}[\mathrm{sgn}(\boldsymbol{w} \cdot \boldsymbol{x}) \neq \mathrm{sgn}(\boldsymbol{w}^\star \cdot \boldsymbol{x})] \leq \epsilon \,,$$
with probability $1 - \delta$, where $\boldsymbol{w}^\star$ is the target normal vector. Since the above misclassification probability with respect to $\mathcal{N}_d$ is directly connected with the angle $\theta(\boldsymbol{w}, \boldsymbol{w}^\star)$, we get that we can control the angle between $\boldsymbol{w}$ and $\boldsymbol{w}^\star$ efficiently. Moreover, in our setting, for a matrix $\boldsymbol{W} \in \mathbb{R}^{k \times d}$, there exist $\binom{k}{2}$ homogeneous halfspaces with normal vectors $\boldsymbol{W}_i - \boldsymbol{W}_j$ and so we can control the angles $\theta(\boldsymbol{W}_i - \boldsymbol{W}_j, \boldsymbol{W}_i^\star - \boldsymbol{W}_j^\star)$. In order to deduce the sample complexity bound of Theorem 5, we show the next lemma which essentially bounds the top-1 misclassification error using the angles of these $O(k^2)$ halfspaces. We apply Lemma 12 with $\boldsymbol{U} = \boldsymbol{W}$ and $\boldsymbol{V} = \boldsymbol{W}^\star$ and so we can take $\epsilon' = \epsilon/(k\sqrt{\log k})$ and invoke the proper learning algorithm of Algorithm 2. This completes the proof. $\qquad\square$

We continue with the proof of our key lemma.

**Lemma 12** (Misclassification Error). *Consider two matrices $U, V \in \mathbb{R}^{k \times d}$ and let $\mathcal{N}_d$ be the standard Gaussian in $d$ dimensions. We have that*

$$\Pr_{\boldsymbol{x} \sim \mathcal{N}_d}[\sigma_1(\boldsymbol{U}\boldsymbol{x}) \neq \sigma_1(\boldsymbol{V}\boldsymbol{x})] \leq c \cdot k \cdot \sqrt{\log k} \cdot \max_{i \neq j} \theta(\boldsymbol{U}_i - \boldsymbol{U}_j, \boldsymbol{V}_i - \boldsymbol{V}_j),$$

*where $c > 0$ is some universal constant.*

*Proof.* We have that

$$\Pr_{\boldsymbol{x} \sim \mathcal{N}_d}[\sigma_1(\boldsymbol{U}\boldsymbol{x}) \neq \sigma_1(\boldsymbol{V}\boldsymbol{x})] = \sum_{i \in [k]} \Pr_{\boldsymbol{x} \sim \mathcal{N}_d}[\sigma_1(\boldsymbol{U}\boldsymbol{x}) = i, \sigma_1(\boldsymbol{V}\boldsymbol{x}) \neq i].$$

We have that $\mathcal{C}_U^{(i)} = \mathbf{1}\{\boldsymbol{x} : \sigma_1(\boldsymbol{U}\boldsymbol{x}) = i\} = \prod_{j \neq i} \mathbf{1}\{(\boldsymbol{U}_i - \boldsymbol{U}_j) \cdot \boldsymbol{x} \geq 0\}$ is the set indicator of a homogeneous polyhedral cone as the intersection of $k-1$ homogeneous halfspaces. Similarly, we consider the cone $\mathcal{C}_V^{(i)} = \{\boldsymbol{x} : \sigma_1(\boldsymbol{V}\boldsymbol{x}) = i\}$. Hence, we have that $\{\boldsymbol{x} : \sigma_1(\boldsymbol{V}\boldsymbol{x}) \neq i\}$ is the complement of a homogeneous polyhedral cone. Let us define $C_U^{(i)} : \mathbb{R}^d \mapsto \{0, 1\}$ and $C_V^{(i)} : \mathbb{R}^d \mapsto \{0, 1\}$ be the associated indicator functions of the two cones. We have that

$$\Pr_{\boldsymbol{x} \sim \mathcal{N}_d}[\sigma_1(\boldsymbol{U}\boldsymbol{x}) = i, \sigma_1(\boldsymbol{V}\boldsymbol{x}) \neq i] = \Pr_{\boldsymbol{x} \sim \mathcal{N}_d}[C_U^{(i)}(\boldsymbol{x}) = 1, C_V^{(i)}(\boldsymbol{x}) = 0].$$

Finally, we have that

$$\mathcal{C}_U^{(i)} \cap \left(\mathcal{C}_V^{(i)}\right)^c = \mathcal{C}_U^{(i)} \setminus \mathcal{C}_V^{(i)} \subseteq \mathcal{C}_U^{(i)} \setminus \mathcal{C}_V^{(i)} \cup \mathcal{C}_V^{(i)} \setminus \mathcal{C}_U^{(i)}.$$

We can hence apply Lemma 13 for the cones $\mathcal{C}_U^{(i)}, \mathcal{C}_V^{(i)}$ for each $i \in [k]$. □

**Lemma 13** (Cone Disagreement). *Let $C_1 : \mathbb{R}^d \mapsto \{0, 1\}$ be the indicator function of the homogeneous polyhedral cone defined by the $k$ unit vectors $\boldsymbol{v}_1, \ldots, \boldsymbol{v}_k \in \mathbb{R}^d$, i.e., $C_1(\boldsymbol{x}) = \prod_{i=1}^k \mathbf{1}\{\boldsymbol{v}_i \cdot \boldsymbol{x} \geq 0\}$. Similarly, define $C_2 : \mathbb{R}^d \mapsto \{0, 1\}$ to be the homogeneous polyhedral cone with normal vectors $\boldsymbol{u}_1, \ldots, \boldsymbol{u}_k$. It holds that*

$$\Pr_{\boldsymbol{x} \sim \mathcal{N}_d}[C_1(\boldsymbol{x}) \neq C_2(\boldsymbol{x})] \leq c\sqrt{\log(k)} \max_{i \in [k]} \theta(\boldsymbol{v}_i, \boldsymbol{u}_i),$$

*where $c > 0$ is some universal constant.*

*Proof.* To simplify notation, denote $\theta = \max_{i \in [k]} \theta(\boldsymbol{v}_i, \boldsymbol{u}_i)$. We first observe that it suffices to prove the upper bound on the probability of $C_1(\boldsymbol{x}) \neq C_2(\boldsymbol{x})$ for sufficiently small values of $\theta$. Indeed, if we have that the bound is true for $\theta$ smaller than some $\theta_0$ we can then form a path of sufficiently large length $N$ (in particular we need $\theta/N \leq \theta_0$) starting from the vectors $\boldsymbol{v}_1, \ldots, \boldsymbol{v}_k$ to the final vectors $\boldsymbol{u}_1, \ldots, \boldsymbol{u}_k$, where at each step we only rotate the vectors by at most $\theta/N \leq \theta_0$. By the triangle inequality, we immediately obtain that the probability that $C_1(\boldsymbol{x}) \neq C_2(\boldsymbol{x})$ is at most equal to the sum of the probabilities of the intermediate steps which is at most $\sum_{i=1}^N c\sqrt{\log(k)}\frac{\theta}{N} = c\sqrt{\log(k)}\theta$. Notice in the above argument the constant $\theta_0$ can be arbitrarily small and may also depend on $k$ and $d$.

We define the indicator of the positive orthant in $k$ dimensions to be $R(\boldsymbol{t}) = \prod_{i=1}^k \mathbf{1}\{t_i \geq 0\}$. Using this notation, we have that the cone indicator can be written as $C_1(\boldsymbol{x}) = R(\boldsymbol{v}_1 \cdot \boldsymbol{x}, \ldots, \boldsymbol{v}_k \cdot \boldsymbol{x}) = R(\boldsymbol{V}\boldsymbol{x})$, where $\boldsymbol{V}$ is the $k \times d$ matrix whose $i$-th row is the vector $\boldsymbol{v}_i$. Moreover, we define the $i$-th face of the cone $R(\boldsymbol{V}\boldsymbol{x})$ to be

$$F_i(\boldsymbol{V}\boldsymbol{x}) = R(\boldsymbol{V}\boldsymbol{x})\,\mathbf{1}\{\boldsymbol{v}_i \cdot \boldsymbol{x} = 0\}.$$

We will first handle the case where only one of the normal vectors $\boldsymbol{v}_i$ changes. We show the following claim.

**Claim 10.** *Let $\boldsymbol{v}_1, \ldots, \boldsymbol{v}_k \in \mathbb{R}^d$ and $\boldsymbol{r} \in \mathbb{R}^d$ with $\theta(\boldsymbol{v}_1, \boldsymbol{r}) \leq \theta$ for some sufficiently small $\theta \in (0, \pi/2)$. It holds that*

$$\Pr_{\boldsymbol{x} \sim \mathcal{N}_d}[R(\boldsymbol{v}_1 \cdot \boldsymbol{x}, \ldots, \boldsymbol{v}_k \cdot \boldsymbol{x}) \neq R(\boldsymbol{r} \cdot \boldsymbol{x}, \boldsymbol{v}_2 \cdot \boldsymbol{x}, \ldots, \boldsymbol{v}_k \cdot \boldsymbol{x})] \leq c \cdot \theta \cdot \Gamma(F_1)\sqrt{\log\left(\frac{1}{\Gamma(F_1)} + 1\right)},$$

*where $F_1$ is the face with $\boldsymbol{v}_1 \cdot \boldsymbol{x} = 0$ of the cone $R(\boldsymbol{V}\boldsymbol{x})$ and $c$ is some universal constant.*

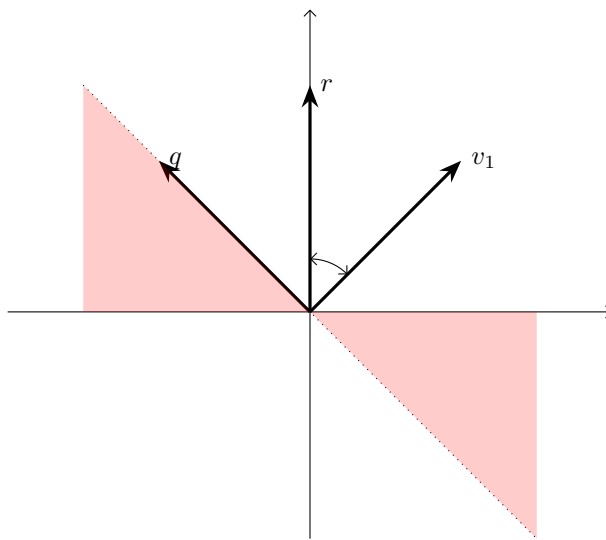

Figure 1: The vectors $r, v_1$ and $q$ and the disagreement region of the halfspaces with normal vectors $r$ and $v_1$.

*Proof.* We have

$$\Pr_{x \sim \mathcal{N}_d} \left[ R(v_1 \cdot x, \ldots, v_k \cdot x) \neq R(r \cdot x, v_2 \cdot x, \ldots, v_k \cdot x) \right]$$

$$= \mathop{\mathbf{E}}_{x \sim \mathcal{N}_d} \left[ |R(v_1 \cdot x, \ldots, v_k \cdot x) - R(r \cdot x, v_2 \cdot x, \ldots, v_k \cdot x)| \right]$$

$$= \mathop{\mathbf{E}}_{x \sim \mathcal{N}_d} \left[ R(v_2 \cdot x, \ldots, v_k \cdot x) \left| \mathbf{1}\{v_1 \cdot x \geq 0\} - \mathbf{1}\{r \cdot x \geq 0\} \right| \right] .$$

We have that $|\mathbf{1}\{v_1 \cdot x \geq 0\} - \mathbf{1}\{r \cdot x \geq 0\}| = \mathbf{1}\{(v_1 \cdot x)(r \cdot x) < 0\}$, i.e., this is the event that the halfspaces $\mathbf{1}\{v_1 \cdot x \geq 0\}$ and $\mathbf{1}\{r \cdot x \geq 0\}$ disagree. Let $q$ be the normalized projection of $r$ onto the orthogonal complement of $v_1$, i.e., $q = \text{proj}_{v_1^\perp} r / \|\text{proj}_{v_1^\perp} r\|_2$. We have that $v_1$ and $q$ is an orthonormal basis of the subspace spanned by the vectors $v_1$ and $r$. We have that $r = \cos\theta(v_1, r) v_1 + \sin\theta(v_1, r) q$. Moreover, we have that the region $(v_1 \cdot x)(r \cdot x) < 0$ is equal to

$$\{0 < v_1 \cdot x < -(q \cdot x) \tan\theta(v_1, r)\} \cup \{-(q \cdot x)\tan\theta(v_1, r) < v_1 \cdot x < 0\} .$$

Thus, we have that the disagreement region $(v_1 \cdot x)(r \cdot x) < 0$ is a subset of the region $\{|v_1 \cdot x| \leq |q \cdot x| \tan\theta(v_1, r)\}$. Since $\tan\theta(v_1, r) \leq \theta$ and we have that $\theta$ is sufficiently small we can also replace the above region by the larger region: $\{|v_1 \cdot x| \leq 2\theta |q \cdot x|\}$. Therefore, we have

$$\mathop{\mathbf{E}}_{x \sim \mathcal{N}_d} \left[ R(v_2 \cdot x, \ldots, v_k \cdot x) \, \mathbf{1}\{(v_1 \cdot x)(r \cdot x) < 0\}\} \right]$$

$$\leq \mathop{\mathbf{E}}_{x \sim \mathcal{N}_d} \left[ R(v_2 \cdot x, \ldots, v_k \cdot x) \, \mathbf{1}\{|v_1 \cdot x| \leq 2\theta |q \cdot x|\} \right] .$$

The derivative of the above expression with respect to $\theta$ is equal to

$$\mathop{\mathbf{E}}_{x \sim \mathcal{N}_d} \left[ R(v_2 \cdot x, \ldots, v_k \cdot x) \, \delta\left( \frac{|v_1 \cdot x|}{2|q \cdot x|} - \theta \right) \right] ,$$

where $\delta(t)$ is the Dirac delta function. At $\theta = 0$ and using the property that $\delta(t/a) = a\delta(t)$, we have that the above derivative is equal to

$$2 \mathop{\mathbf{E}}_{x \sim \mathcal{N}_d} \left[ R(v_2 \cdot x, \ldots, v_k \cdot x) \, |q \cdot x| \, \delta(|v_1 \cdot x|) \right] .$$

Notice that, if we did not have the term $|q \cdot x|$, the above expression would be exactly equal to two times the Gaussian surface area of the face with $v_1 \cdot x = 0$, i.e., it would be equal to $2\Gamma(F_1)$. We now show that this extra term of $|q \cdot x|$ can only increase the above surface integral by at most a

logarithmic factor. We have that

$$\mathop{\mathbf{E}}_{\boldsymbol{x}\sim\mathcal{N}_d}\left[R(\boldsymbol{v}_2\cdot\boldsymbol{x},\ldots,\boldsymbol{v}_k\cdot\boldsymbol{x})\,|\boldsymbol{q}\cdot\boldsymbol{x}|\,\delta(|\boldsymbol{v}_1\cdot\boldsymbol{x}|)\right]=\int_{\boldsymbol{x}\in F_1}\phi_d(\boldsymbol{x})|\boldsymbol{q}\cdot\boldsymbol{x}|d\mu(\boldsymbol{x})$$

$$\leq\int_{\boldsymbol{x}\in F_1}\phi_d(\boldsymbol{x})|\boldsymbol{q}\cdot\boldsymbol{x}|\mathbf{1}\{|\boldsymbol{q}\cdot\boldsymbol{x}|\leq\xi\}d\mu(\boldsymbol{x})+\int_{\boldsymbol{x}\in F_1}\phi_d(\boldsymbol{x})|\boldsymbol{q}\cdot\boldsymbol{x}|\mathbf{1}\{|\boldsymbol{q}\cdot\boldsymbol{x}|\geq\xi\}d\mu(\boldsymbol{x})$$

$$\leq\xi\int_{\boldsymbol{x}\in F_1}\phi_d(\boldsymbol{x})d\mu(\boldsymbol{x})+\int_{\boldsymbol{x}\in F_1}\phi_d(\boldsymbol{x})|\boldsymbol{q}\cdot\boldsymbol{x}|\mathbf{1}\{|\boldsymbol{q}\cdot\boldsymbol{x}|\geq\xi\}d\mu(\boldsymbol{x})\,,$$

where $d\mu(\boldsymbol{x})$ is the standard surface measure in $\mathbb{R}^d$. The first term above is exactly equal to the Gaussian surface area of the face $F_1$. To bound from above the second term we can use the fact that the face $F_1$ is a subset of the hyperplane $\boldsymbol{v}_1\cdot\boldsymbol{x}=0$, i.e., it holds that $F_1\subseteq\{\boldsymbol{x}:|\boldsymbol{v}_1\cdot\boldsymbol{x}|=0\}$. To simplify notation we may assume that $\boldsymbol{v}_1=\boldsymbol{e}_1$ and $\boldsymbol{q}=\boldsymbol{e}_2$ (recall that $\boldsymbol{v}_1$ and $\boldsymbol{q}$ are orthogonal unit vectors), and in this case we obtain

$$\int_{\boldsymbol{x}\in F_1}\phi_d(\boldsymbol{x})|\boldsymbol{q}\cdot\boldsymbol{x}|\mathbf{1}\{|\boldsymbol{q}\cdot\boldsymbol{x}|\geq\xi\}d\mu(\boldsymbol{x})\leq\int_{x_1=0}\phi_d(\boldsymbol{x})|x_2|\mathbf{1}\{|x_2|\geq\xi\}d\mu(\boldsymbol{x})$$

$$=\frac{1}{\sqrt{2\pi}}\int_{-\infty}^{+\infty}|x_2|\mathbf{1}\{|x_2|\geq\xi\}\frac{e^{-x_2^2/2}}{\sqrt{2\pi}}dx_2$$

$$=\frac{1}{\pi}e^{-\xi^2/2}\,.$$

Combining the above bounds we obtain that the derivative with respect to $\theta$ of the expression $\mathbf{E}_{\boldsymbol{x}\sim\mathcal{N}_d}\left[R(\boldsymbol{v}_2\cdot\boldsymbol{x},\ldots,\boldsymbol{v}_k\cdot\boldsymbol{x})\,\mathbf{1}\{|\boldsymbol{v}_1\cdot\boldsymbol{x}|\leq2\theta|\boldsymbol{q}\cdot\boldsymbol{x}|\}\right]$ is equal to

$$\frac{d}{d\theta}\left(\mathop{\mathbf{E}}_{\boldsymbol{x}\sim\mathcal{N}_d}\left[R(\boldsymbol{v}_2\cdot\boldsymbol{x},\ldots,\boldsymbol{v}_k\cdot\boldsymbol{x})\,\mathbf{1}\{|\boldsymbol{v}_1\cdot\boldsymbol{x}|\leq2\theta|\boldsymbol{q}\cdot\boldsymbol{x}|\}\right]\right)\Big|_{\theta=0}\leq2\xi\Gamma(F_1)+\frac{2e^{-\xi^2/2}}{\pi}\,.$$

By picking $\xi=\sqrt{2\log(1+1/\Gamma(F_1))}$, the result follows since up to introducing $o(\theta)$ error we can bound the term $\mathbf{Pr}_{\boldsymbol{x}\sim\mathcal{N}_d}\left[R(\boldsymbol{v}_1\cdot\boldsymbol{x},\ldots,\boldsymbol{v}_k\cdot\boldsymbol{x})\neq R(\boldsymbol{r}\cdot\boldsymbol{x},\boldsymbol{v}_2\cdot\boldsymbol{x},\ldots,\boldsymbol{v}_k\cdot\boldsymbol{x})\right]$ by its derivative with respect to $\theta$ (evaluated at 0) times $\theta$. $\qquad\square$

We can complete the proof of Lemma 13 using Claim 10. In order to bound the disagreement of the cones $C_1$ and $C_2$ we can start from $C_1$ and change one of its vectors at a time so that we can use Claim 10 that can handle this case. For example, at the first step, we can swap $\boldsymbol{v}_1$ for $\boldsymbol{u}_1$ and use the triangle inequality to obtain that

$$\mathop{\mathbf{Pr}}_{\boldsymbol{x}\sim\mathcal{N}_d}\left[C_1(\boldsymbol{x})\neq C_2(\boldsymbol{x})\right]\leq\mathop{\mathbf{Pr}}_{\boldsymbol{x}\sim\mathcal{N}_d}\left[R(\boldsymbol{v}_1\cdot\boldsymbol{x},\ldots,\boldsymbol{v}_k\cdot\boldsymbol{x})\neq R(\boldsymbol{u}_1\cdot\boldsymbol{x},\boldsymbol{v}_2\cdot\boldsymbol{x}\ldots,\boldsymbol{v}_k\cdot\boldsymbol{x})\right]$$

$$+\mathop{\mathbf{Pr}}_{\boldsymbol{x}\sim\mathcal{N}_d}\left[R(\boldsymbol{u}_1\cdot\boldsymbol{x},\boldsymbol{v}_2\cdot\boldsymbol{x},\ldots,\boldsymbol{v}_k\cdot\boldsymbol{x})\neq R(\boldsymbol{u}_1\cdot\boldsymbol{x},\boldsymbol{u}_2\cdot\boldsymbol{x}\ldots,\boldsymbol{u}_k\cdot\boldsymbol{x})\right]$$

$$\leq c\cdot\theta\,\Gamma(F_1)\sqrt{\log(1/\Gamma(F_1)+1)}$$

$$+\mathop{\mathbf{Pr}}_{\boldsymbol{x}\sim\mathcal{N}_d}\left[R(\boldsymbol{u}_1\cdot\boldsymbol{x},\boldsymbol{v}_2\cdot\boldsymbol{x},\ldots,\boldsymbol{v}_k\cdot\boldsymbol{x})\neq R(\boldsymbol{u}_1\cdot\boldsymbol{x},\boldsymbol{u}_2\cdot\boldsymbol{x}\ldots,\boldsymbol{u}_k\cdot\boldsymbol{x})\right]\,,$$

where $F_1=F_1(\boldsymbol{V}\boldsymbol{x})$ is the face with $\boldsymbol{v}_1\cdot\boldsymbol{x}=0$ of the cone $C_1$. Notice that we have replaced $\boldsymbol{v}_1$ by $\boldsymbol{u}_1$ in the above bound. Our plan is to use the triangle inequality and continue replacing the vectors of $C_1$ by the vectors of $C_2$ sequentially. To make this formal we define the matrix $\boldsymbol{A}^{(i)}\in\mathbb{R}^{k\times d}$ whose first $i-1$ rows are the vectors $\boldsymbol{u}_1,\ldots,\boldsymbol{u}_{i-1}$ and its last $k-i+1$ rows are the vectors $\boldsymbol{v}_i,\ldots,\boldsymbol{v}_k$, i.e.,

$$\boldsymbol{A}_j^{(i)}=\begin{cases}\boldsymbol{u}_j & \text{if}\quad 1\leq j\leq i-1,\\\boldsymbol{v}_j & \text{if}\quad i\leq j\leq k\,.\end{cases}$$

Notice that $\boldsymbol{A}^{(1)}=\boldsymbol{V}$ and $\boldsymbol{A}^{(k+1)}=\boldsymbol{U}$. Using the triangle inequality we obtain that

$$\mathop{\mathbf{Pr}}_{\boldsymbol{x}\sim\mathcal{N}_d}\left[C_1(\boldsymbol{x})\neq C_2(\boldsymbol{x})\right]\leq\sum_{i=1}^{k}\mathop{\mathbf{Pr}}_{\boldsymbol{x}\sim\mathcal{N}_d}\left[R(\boldsymbol{A}^{(i)}\boldsymbol{x})\neq R(\boldsymbol{A}^{(i+1)}\boldsymbol{x})\right].$$

Since the matrices $A^{(i)}$ and $A^{(i+1)}$ only differ on one row, we can use Claim 10 to obtain the following bound:

$$\Pr_{x \sim \mathcal{N}_d}[C_1(x) \neq C_2(x)] \leq c \cdot \theta \cdot \sum_{i=1}^{k} \Gamma(F_i(A^{(i)}x)) \sqrt{\frac{1}{\Gamma(F_i(A^{(i)}x))} + 1} \,.$$

We now observe that the Gaussian surface area $\Gamma(F_i(A^{(i)}x))$ is a continuous function of the matrix $A^{(i)}$. By flattening the matrix $A^{(i)}$ (since it is isomorphic to a vector $z \in \mathbb{R}^{n^2}$) and letting $S_z$ be the induced surface $\{x : R(A^{(i)}x) = 1 \wedge v_i \cdot x = 0\}$, it suffices to show that

$$\lim_{w \to z} \int \phi_n(x) \mathbf{1}\{x \in S_w\} d\mu(x) = \int \phi_n(x) \mathbf{1}\{x \in S_z\} d\mu(x) \,,$$

by the smoothness of the surface $S_z$. Consider a sequence of functions $(g_m)$ and vectors $(w_m)$ so that $g_m(x) = \phi_n(x) \mathbf{1}\{x \in S_{w_m}\}$ and $\lim_{m \to \infty} w_m = z$. Note that $|g_m(x)| \leq 1$ everywhere. Hence, by the dominated convergence theorem, we have that

$$\lim_{m \to \infty} \int g_m(x) d\mu(x) = \int \lim_{m \to \infty} g_m(x) d\mu(x) = \int \phi_n(x) \lim_{m \to \infty} \mathbf{1}\{x \in S_{w_m}\} d\mu(x) \,.$$

Since the sequence consists of smooth surfaces, we have that $\lim_{m \to \infty} \mathbf{1}\{x \in S_{w_m}\} = \mathbf{1}\{x \in S_z\}$ and so the Gaussian surface area is continuous with respect to the matrix $A^{(i)}$ for any $i \in [k]$.

Also, as $\theta \to 0$, we have that $A^{(i)} \to V$. This is because the sequence of matrices $A^{(i)}$ depends only on the vectors $u_j$ and $v_j$ for $j \in [k]$ and the following two properties hold true: $\theta = \max_{j \in [k]} \theta(v_j, u_j)$ and all the vectors are unit. Hence, as $\theta$ tends to zero, they tend to become the same vectors and so any matrix $A^{(i)}$ tends to become $V$. Therefore, taking this limit we obtain that for $\theta \to 0$ it holds that

$$\lim_{\theta \to 0} \frac{\Pr_{x \sim \mathcal{N}_d}[C_1(x) \neq C_2(x)]}{\theta} \leq c \cdot \sum_{i=1}^{k} \Gamma(F_i(Vx)) \sqrt{\log\left(1/\Gamma(F_i(Vx)) + 1\right)} \,. \tag{1}$$

We will now use the following lemma that shows that the surface area of any homogeneous polyhedral cone is independent of the number of faces $k$ and in fact is at most 1 for all $k$.

**Lemma 14** (Gaussian Surface Area of Homogeneous Cones [Naz03])**.** *Let $C$ be a cone with apex at the origin (i.e., an intersection of arbitrarily many halfspaces all of whose boundaries contain the origin). Then $C$ has Gaussian surface area $\Gamma(C)$ at most 1.*

Using Lemma 14 we obtain that $\sum_{i=1}^{k} \Gamma(F_i(Vx)) \leq 1$. Next, we observe that, when the positive numbers $a_1, \ldots, a_k$ satisfy $\sum_{i=1}^{k} a_i \leq 1$, it holds that $\sum_{i=1}^{k} a_i \sqrt{\log(1/a_i)} \leq \sqrt{\sum_{i=1}^{k} a_i \log(1/a_i)} \leq \sqrt{\log(k)}$ (using the fact that the uniform distribution maximizes the entropy). Using this fact and Equation (1), we obtain

$$\lim_{\theta \to 0} \frac{\Pr_{x \sim \mathcal{N}_d}[C_1(x) \neq C_2(x)]}{\theta} \leq c\sqrt{\log(k)} \,.$$

Thus, we have shown that, for sufficiently small $\theta$, it holds that $\Pr_{x \sim \mathcal{N}_d}[C_1(x) \neq C_2(x)] \leq c\sqrt{\log(k)}\theta$, but, as we discussed in the start of the proof, the general bound follows directly from the bound for sufficiently small values of $\theta > 0$. $\qquad\square$

## C  Learning in Top-$r$ Disagreement from Label Rankings

We prove the next result which corresponds to a proper learning algorithm for LSF in the presence of bounded noise with respect to the top-$r$ disagreement.

**Theorem 6** (Proper Top-$r$ Learning Algorithm)**.** *Fix $\eta \in [0, 1/2)$, $r \in [k]$ and $\epsilon, \delta \in (0, 1)$. Let $\mathcal{D}$ be an $\eta$-noisy linear label ranking distribution satisfying the assumptions of Definition 2. There exists an algorithm that draws $N = \widetilde{O}\left(\frac{d\,rk}{\epsilon(1-2\eta)^6} \log(1/\delta)\right)$ samples from $\mathcal{D}$, runs in $\mathrm{poly}(N)$ time and, with probability at least $1 - \delta$, outputs a Linear Sorting function $h : \mathbb{R}^d \to \mathbb{S}_k$ that is $\epsilon$-close in top-$r$ disagreement to the target.*

The main result of this section is the next lemma, which directly implies the above theorem (using the same steps as the proof of Theorem 5).

**Lemma 15** (Top-$r$ Misclassification). *Let $r \in [k]$. Consider two matrices $\boldsymbol{U}, \boldsymbol{V} \in \mathbb{R}^{k \times d}$ and let $\mathcal{N}_d$ be the standard Gaussian in $d$ dimensions. We have that*

$$\Pr_{\boldsymbol{x} \sim \mathcal{N}_d}[\sigma_{1..r}(\boldsymbol{U}\boldsymbol{x}) \neq \sigma_{1..r}(\boldsymbol{V}\boldsymbol{x})] \leq c \cdot k \cdot r \cdot \sqrt{\log(kr)} \cdot \max_{i \neq j} \theta(\boldsymbol{U}_i - \boldsymbol{U}_j, \boldsymbol{V}_i - \boldsymbol{V}_j),$$

*where $c > 0$ is some universal constant.*

*Proof.* Let us set $\sigma_{1..r}(\boldsymbol{W}\boldsymbol{x})$ denote the ordering of the top-$r$ alternatives in the ranking $\sigma(\boldsymbol{W}\boldsymbol{x})$. Moreover, recall that $\sigma_\ell(\boldsymbol{W}\boldsymbol{x})$ denotes the alternative in the $\ell$-th position of the ranking $\sigma(\boldsymbol{W}\boldsymbol{x})$. For two matrices $\boldsymbol{U}, \boldsymbol{V} \in \mathbb{R}^{k \times d}$, we have that

$$\Pr_{\boldsymbol{x} \sim \mathcal{N}_d}[\sigma_{1..r}(\boldsymbol{U}\boldsymbol{x}) \neq \sigma_{1..r}(\boldsymbol{V}\boldsymbol{x})] = \sum_{j=1}^{k} \Pr_{\boldsymbol{x} \sim \mathcal{N}_d}\left[\bigcup_{\ell=1}^{r}\{j = \sigma_\ell(\boldsymbol{U}\boldsymbol{x}), j \neq \sigma_\ell(\boldsymbol{V}\boldsymbol{x})\}\right].$$

The first step is to understand the geometry of the set $\bigcup_{\ell=1}^{r}\{\boldsymbol{x} : j = \sigma_\ell(\boldsymbol{U}\boldsymbol{x})\} = \{\boldsymbol{x} : j \in \sigma_{1..r}(\boldsymbol{U}\boldsymbol{x})\}$ for $j \in [k]$. We have that this set is equal to

$$\mathcal{T}_{\boldsymbol{U}}^{(j)} = \bigcup_{S \subseteq [k]:|S| \leq r-1} \bigcap_{i \in S}\{\boldsymbol{x} : (\boldsymbol{U}_i - \boldsymbol{U}_j) \cdot \boldsymbol{x} \geq 0\} \cap \bigcap_{i \notin S}\{\boldsymbol{x} : (\boldsymbol{U}_i - \boldsymbol{U}_j) \cdot \boldsymbol{x} \leq 0\}.$$

In words, $\mathcal{T}_{\boldsymbol{U}}^{(j)}$ iterates over any possible collection of alternatives that can win the element $j$ (they lie in the set of top elements $S$) and the remaining elements lose when compared with $j$ (they lie in the complement set $[k] \setminus S$). Overloading the notation, let us define the mapping $T(\boldsymbol{t}) = T(t_1, ..., t_k) = \sum_{S \subseteq [k]:|S| \leq r-1} \prod_{i \in S} \mathbf{1}\{t_i \geq 0\} \prod_{i \notin S} \mathbf{1}\{t_i \leq 0\}$. Using this mapping, we can define the indicator of the set $\mathcal{T}_{\boldsymbol{U}}^{(j)}$ as $T((\boldsymbol{U}_1 - \boldsymbol{U}_j) \cdot \boldsymbol{x}, \ldots, (\boldsymbol{U}_k - \boldsymbol{U}_j) \cdot \boldsymbol{x})$. The top-$r$ disagreement $\Pr_{\boldsymbol{x} \sim \mathcal{N}_d}[j \in \sigma_{1..r}(\boldsymbol{U}\boldsymbol{x}), j \notin \sigma_{1..r}(\boldsymbol{V}\boldsymbol{x})]$ is equal to:

$$\Pr_{\boldsymbol{x} \sim \mathcal{N}_d}[T((\boldsymbol{U}_1 - \boldsymbol{U}_j) \cdot \boldsymbol{x}, ..., (\boldsymbol{U}_k - \boldsymbol{U}_j) \cdot \boldsymbol{x}) = 1, T((\boldsymbol{V}_1 - \boldsymbol{V}_j) \cdot \boldsymbol{x}, ..., (\boldsymbol{V}_k - \boldsymbol{V}_j) \cdot \boldsymbol{x}) = 0].$$

So we have that

$$\Pr_{\boldsymbol{x} \sim \mathcal{N}_d}[\sigma_{1..r}(\boldsymbol{U}\boldsymbol{x}) \neq \sigma_{1..r}(\boldsymbol{V}\boldsymbol{x})] = \sum_{j=1}^{k} \Pr_{\boldsymbol{x} \sim \mathcal{N}_d}[T_j(\boldsymbol{U}\boldsymbol{x}) = 1, T_j(\boldsymbol{V}\boldsymbol{x}) = 0] \leq \sum_{j=1}^{k} \Pr_{\boldsymbol{x} \sim \mathcal{N}_d}[T_j(\boldsymbol{U}\boldsymbol{x}) \neq T_j(\boldsymbol{V}\boldsymbol{x})].$$

In order to show the desired bound, it suffices to prove the following two lemmas.

**Lemma 16** (Disagreement Region). *Consider a positive integer $r \leq k$. Fix $j \in [k]$ and let $\theta = \max_{i \in [k]} \theta(\boldsymbol{U}_i - \boldsymbol{U}_j, \boldsymbol{V}_i - \boldsymbol{V}_j)$. Then it holds that*

$$\lim_{\theta \to 0} \frac{\Pr_{\boldsymbol{x} \sim \mathcal{N}_d}[T_j(\boldsymbol{U}\boldsymbol{x}) \neq T_j(\boldsymbol{V}\boldsymbol{x})]}{\theta} \leq c \cdot \sum_{i \in [k]} \Gamma(F_i^j) \sqrt{\log\left(\frac{1}{\Gamma(F_i^j)} + 1\right)},$$

*where $c > 0$ is some constant and $F_i^j$ is the surface $\{\boldsymbol{x} : j \in \sigma_{1..r}(\boldsymbol{V}\boldsymbol{x})\} \cap \{\boldsymbol{x} : \boldsymbol{V}_i \cdot \boldsymbol{x} = \boldsymbol{V}_j \cdot \boldsymbol{x}\}$ for the matrix $\boldsymbol{V} \in \mathbb{R}^{k \times d}$.*

and,

**Lemma 17.** *Let $F_i^j, r, k$ as in the previous lemma. It holds that*

$$\sum_{i \in [k]} \sum_{j \in [k]} \Gamma(F_i^j) \leq 2kr.$$

Applying these two lemmas with $\theta = \max_{i \neq j} \theta(\boldsymbol{U}_i - \boldsymbol{U}_j, \boldsymbol{V}_i - \boldsymbol{V}_j)$, we get that

$$Z := \lim_{\theta \to 0} \frac{\sum_{j \in [k]} \Pr_{\boldsymbol{x} \sim \mathcal{N}_d}[T_j(\boldsymbol{U}\boldsymbol{x}) \neq T_j(\boldsymbol{V}\boldsymbol{x})]}{\theta} \leq c \cdot \sum_{j \in [k]} \sum_{i \in [k]} \Gamma(F_i^j) \sqrt{\log\left(\frac{1}{\Gamma(F_i^j)} + 1\right)}.$$

Let us set $\Gamma'(F_i^j) = \Gamma(F_i^j)/(2kr)$. Then we have that

$$Z \leq 2ckr \cdot \sum_{j \in [k]} \sum_{i \in [k]} \Gamma'(F_i^j) \sqrt{\log\left(\frac{1}{2kr \cdot \Gamma'(F_i^j)} + 1\right)}.$$

It suffices to bound the quantity

$$\sum_{j \in [k]} \sum_{i \in [k]} \Gamma'(F_i^j) \sqrt{\log\left(\frac{1}{\Gamma'(F_i^j)} + 1\right)} = O\left(kr\sqrt{\log(kr)}\right),$$

where we used a similar "entropy-like" inequality as we did in the top-1 case. This yields (by recalling that it is sufficient to consider only the case of arbitrarily small angles, as in the top-1 case) that

$$\Pr_{\boldsymbol{x} \sim \mathcal{N}_d}[\sigma_{1..r}(\boldsymbol{U}\boldsymbol{x}) \neq \sigma_{1..r}(\boldsymbol{V}\boldsymbol{x})] \leq c\,rk\,\sqrt{\log(kr)} \cdot \max_{i \neq j} \theta(\boldsymbol{U}_i - \boldsymbol{U}_j, \boldsymbol{V}_i - \boldsymbol{V}_j),$$

for some universal constant $c$. $\qquad\square$

## C.1 The proof of Lemma 16

We proceed with the proof of the key lemma concerning the disagreement region. We first show the following claim where we only change a single vector. Recall that

$$T(\boldsymbol{V}\boldsymbol{x}) = \sum_{S:|S| \leq r-1} \prod_{i \in S} \mathbf{1}\{\boldsymbol{v}_i \cdot \boldsymbol{x} \geq 0\} \prod_{i \notin S} \mathbf{1}\{\boldsymbol{v}_i \cdot \boldsymbol{x} \leq 0\}.$$

We will be interested in the surface $F_1 := F_1(\boldsymbol{V}\boldsymbol{x}) = T(\boldsymbol{V}\boldsymbol{x})\mathbf{1}\{\boldsymbol{v}_1 \cdot \boldsymbol{x} = 0\}$.

**Claim 11.** *Let $\boldsymbol{v}_1, \ldots, \boldsymbol{v}_k \in \mathbb{R}^d$ and $\boldsymbol{r} \in \mathbb{R}^d$ with $\theta(\boldsymbol{v}_1, \boldsymbol{r}) \leq \theta$ for some sufficiently small $\theta \in (0, \pi/2)$. It holds that*

$$\Pr_{\boldsymbol{x} \sim \mathcal{N}_d}[T(\boldsymbol{v}_1 \cdot \boldsymbol{x}, \ldots, \boldsymbol{v}_k \cdot \boldsymbol{x}) \neq T(\boldsymbol{r} \cdot \boldsymbol{x}, \boldsymbol{v}_2 \cdot \boldsymbol{x}, \ldots, \boldsymbol{v}_k \cdot \boldsymbol{x})] \leq c \cdot \theta \cdot \Gamma(F_1)\sqrt{\log\left(\frac{1}{\Gamma(F_1)} + 1\right)},$$

*where $F_1$ is the surface $T(\boldsymbol{V}\boldsymbol{x}) \cap \{\boldsymbol{x} : \boldsymbol{v}_1 \cdot \boldsymbol{x} = 0\}$ and $c$ is some universal constant.*

*Proof.* We first decompose the sum of $T(\boldsymbol{V}\boldsymbol{x})$ depending on whether $1 \in S$ or not. Hence, we have that $T(\boldsymbol{v}_1 \cdot \boldsymbol{x}, \ldots, \boldsymbol{v}_k \cdot \boldsymbol{x}) = T^+(\boldsymbol{v}_1 \cdot \boldsymbol{x}, \ldots, \boldsymbol{v}_k \cdot \boldsymbol{x}) + T^-(\boldsymbol{v}_1 \cdot \boldsymbol{x}, \ldots, \boldsymbol{v}_k \cdot \boldsymbol{x})$ where

$$T^+(\boldsymbol{v}_1 \cdot \boldsymbol{x}, \ldots, \boldsymbol{v}_k \cdot \boldsymbol{x}) = \sum_{S \subseteq [k]:|S| \leq r-1, 1 \in S} \prod_{i \in S} \mathbf{1}\{\boldsymbol{v}_i \cdot \boldsymbol{x} \geq 0\} \prod_{i \notin S} \mathbf{1}\{\boldsymbol{v}_i \cdot \boldsymbol{x} \leq 0\}$$

$$= \sum_{S \subseteq [k]:|S| \leq r-1, 1 \in S} \mathbf{1}\{\boldsymbol{v}_1 \cdot \boldsymbol{x} \geq 0\} \cdot \prod_{i \in S \setminus \{1\}} \mathbf{1}\{\boldsymbol{v}_i \cdot \boldsymbol{x} \geq 0\} \prod_{i \notin S} \mathbf{1}\{\boldsymbol{v}_i \cdot \boldsymbol{x} \leq 0\}$$

$$= \mathbf{1}\{\boldsymbol{v}_1 \cdot \boldsymbol{x} \geq 0\} \cdot \sum_{S \subseteq [k]:|S| \leq r-1, 1 \in S} \prod_{i \in S \setminus \{1\}} \mathbf{1}\{\boldsymbol{v}_i \cdot \boldsymbol{x} \geq 0\} \prod_{i \notin S} \mathbf{1}\{\boldsymbol{v}_i \cdot \boldsymbol{x} \leq 0\}$$

$$=: \mathbf{1}\{\boldsymbol{v}_1 \cdot \boldsymbol{x} \geq 0\} \cdot G^+(\boldsymbol{v}_2 \cdot \boldsymbol{x}, \ldots, \boldsymbol{v}_k \cdot \boldsymbol{x}),$$

and similarly

$$T^-(\boldsymbol{v}_1 \cdot \boldsymbol{x}, \ldots, \boldsymbol{v}_k \cdot \boldsymbol{x}) = \mathbf{1}\{\boldsymbol{v}_1 \cdot \boldsymbol{x} \leq 0\} \cdot \sum_{S \subseteq [k]:|S| \leq r-1, 1 \notin S} \prod_{i \in S} \mathbf{1}\{\boldsymbol{v}_i \cdot \boldsymbol{x} \geq 0\} \prod_{i \notin S \setminus \{1\}} \mathbf{1}\{\boldsymbol{v}_i \cdot \boldsymbol{x} \leq 0\}$$

$$=: \mathbf{1}\{\boldsymbol{v}_1 \cdot \boldsymbol{x} \leq 0\} \cdot G^-(\boldsymbol{v}_2 \cdot \boldsymbol{x}, \ldots, \boldsymbol{v}_k \cdot \boldsymbol{x}).$$

Notice that the indicator $G^s$ does not depend on the alternative 1 for $s \in \{-, +\}$. Since $T : \mathbb{R}^k \rightarrow \{0, 1\}$, we have that

$$\Pr_{\boldsymbol{x} \sim \mathcal{N}_d}[T(\boldsymbol{v}_1 \cdot \boldsymbol{x}, \ldots, \boldsymbol{v}_k \cdot \boldsymbol{x}) \neq T(\boldsymbol{r} \cdot \boldsymbol{x}, \boldsymbol{v}_2 \cdot \boldsymbol{x}, \ldots, \boldsymbol{v}_k \cdot \boldsymbol{x})]$$

$$= \mathbb{E}_{\boldsymbol{x} \sim \mathcal{N}_d}[|T(\boldsymbol{v}_1 \cdot \boldsymbol{x}, \ldots, \boldsymbol{v}_k \cdot \boldsymbol{x}) - T(\boldsymbol{r} \cdot \boldsymbol{x}, \boldsymbol{v}_2 \cdot \boldsymbol{x}, \ldots, \boldsymbol{v}_k \cdot \boldsymbol{x})|]$$

$$\leq \sum_{s \in \{-, +\}} \mathbb{E}_{\boldsymbol{x} \sim \mathcal{N}_d}[|T^s(\boldsymbol{v}_1 \cdot \boldsymbol{x}, \ldots, \boldsymbol{v}_k \cdot \boldsymbol{x}) - T^s(\boldsymbol{r} \cdot \boldsymbol{x}, \boldsymbol{v}_2 \cdot \boldsymbol{x}, \ldots, \boldsymbol{v}_k \cdot \boldsymbol{x})|]$$

$$= \sum_{s \in \{-, +\}} \mathbb{E}_{\boldsymbol{x} \sim \mathcal{N}_d}[G^s(\boldsymbol{v}_2 \cdot \boldsymbol{x}, \ldots, \boldsymbol{v}_k \cdot \boldsymbol{x}) \cdot |\mathbf{1}\{s \cdot \boldsymbol{v}_1 \cdot \boldsymbol{x} \geq 0\} - \mathbf{1}\{s \cdot \boldsymbol{r} \cdot \boldsymbol{x} \geq 0\}|].$$

Let us focus on the case $s = +$. The difference between the two indicators in the last line of the above equation corresponds to the event that the halfspaces $\mathbf{1}\{v_1 \cdot x \geq 0\}$ and $\mathbf{1}\{r \cdot x \geq 0\}$ disagree. Hence, we have that $|\mathbf{1}\{v_1 \cdot x \geq 0\} - \mathbf{1}\{r \cdot x \geq 0\}| = \mathbf{1}\{(v_1 \cdot x)(r \cdot x) < 0\}$. Note that the above indicator depends on both $v_1$ and $r$. We would like to work only with one of these two vectors. To this end, let us introduce $q$, the normalized projection of $r$ onto the orthogonal complement of $v_1$, i.e., $q = \mathrm{proj}_{v_1^\perp} r / \|\mathrm{proj}_{v_1^\perp} r\|_2$. We have that $v_1$ and $q$ is an orthonormal basis of the subspace spanned by the vectors $v_1$ and $r$. Notice that $r = \cos\theta(v_1, r)v_1 + \sin\theta(v_1, r)q$, by the construction of $q$. Our goal is to understand the structure of the region $(v_1 \cdot x)(r \cdot x) < 0$. This set is equal to

$$\{0 < v_1 \cdot x < -(q \cdot x)\tan\theta(v_1, r)\} \cup \{-(q \cdot x)\tan\theta(v_1, r) < v_1 \cdot x < 0\}\,.$$

To see this, we have that $(v_1 \cdot x)(r \cdot x) = (v_1 \cdot x)(\cos\theta(v_1, r)v_1 \cdot x + \sin\theta(v_1, r)q \cdot x)$. This quantity must be negative. The left-hand set considers the case where $v_1 \cdot x > 0$ and so $\tan\theta(v_1, r)(q \cdot x) < -v_1 \cdot x$. We obtain the right-hand set in a similar way. Thus, we have that the disagreement region $(v_1 \cdot x)(r \cdot x) < 0$ is a subset of the region $\{|v_1 \cdot x| \leq |q \cdot x|\tan\theta(v_1, r)\}$. Since $\tan\theta(v_1, r) \leq \theta$ and we have that $\theta$ is sufficiently small we can also replace the above region by the larger region: $\{|v_1 \cdot x| \leq 2\theta|q \cdot x|\}$. Therefore, we have

$$\underset{x \sim \mathcal{N}_d}{\mathbf{E}}\left[G^+(v_2 \cdot x, \ldots, v_k \cdot x)\,\mathbf{1}\{(v_1 \cdot x)(r \cdot x) < 0\}\}\right]$$
$$\leq \underset{x \sim \mathcal{N}_d}{\mathbf{E}}\left[G^+(v_2 \cdot x, \ldots, v_k \cdot x)\,\mathbf{1}\{|v_1 \cdot x| \leq 2\theta|q \cdot x|\}\}\right]\,.$$

From this point, the proof goes as in the top-1 case. In total, we will get that

$$\underset{x \sim \mathcal{N}_d}{\mathbf{Pr}}[T(v_1 \cdot x, \ldots, v_k \cdot x) \neq T(r \cdot x, v_2 \cdot x, \ldots, v_k \cdot x)]$$
$$= \underset{x \sim \mathcal{N}_d}{\mathbf{E}}\left[(G^+(v_2 \cdot x, \ldots, v_k \cdot x) + G^-(v_2 \cdot x, \ldots, v_k \cdot x))\,|q \cdot x|\,\delta(|v_1 \cdot x|)\right]$$
$$\leq 2\int_{x \in F_1} \phi_d(x)|q \cdot x|d\mu(x)$$
$$\leq 2\int_{x \in F_1} \phi_d(x)|q \cdot x|\mathbf{1}\{|q \cdot x| \leq \xi\}d\mu(x) + 2\int_{x \in F_1} \phi_d(x)|q \cdot x|\mathbf{1}\{|q \cdot x| \geq \xi\}d\mu(x)$$
$$\leq 2\xi\int_{x \in F_1} \phi_d(x)d\mu(x) + 2\int_{x \in F_1} \phi_d(x)|q \cdot x|\mathbf{1}\{|q \cdot x| \geq \xi\}d\mu(x)\,,$$

where $d\mu(x)$ is the standard surface measure in $\mathbb{R}^d$. Let us explain the first inequality above. Note that the space induced by $G^-(v_2 \cdot x, \ldots, v_k \cdot x)$ contains the space induced by $G^+(v_2 \cdot x, \ldots, v_k \cdot x)$. Hence, in the integration, we can integrate over the surface $F_1 = T(Vx) \cap \mathbf{1}\{x : v_1 \cdot x = 0\}$ twice. Essentially, this surface corresponds to $\mathbf{1}\{v_1 \cdot x = 0\} \cdot \sum_{S \subseteq [k]\setminus\{1\}:|S|\leq r-1} \prod_{i \in S} \mathbf{1}\{v_i \cdot x \geq 0\} \prod_{i \notin S} \mathbf{1}\{v_i \cdot x \leq 0\}$. Applying the steps of the top-1 case, we can obtain the desired bound in terms of the Gaussian surface area of $F_1$. $\square$

Next, for fixed $j \in [k]$, we can apply the above claim sequentially (as we did in the end of the top-1 case) to get

$$\lim_{\theta \to 0} \frac{\mathbf{Pr}_{x \sim \mathcal{N}_d}[T_j(Ux) \neq T_j(Vx)]}{\theta} \leq c \cdot \sum_{i \in [k]} \Gamma(F_i^j)\sqrt{\log\left(\frac{1}{\Gamma(F_i^j)} + 1\right)}\,,$$

for some small constant $c > 0$.

## C.2   The proof of Lemma 17

Using the above result, we get that it suffices to control the value $\Gamma(F_i^j)$, where $F_i^j$ is the surface of $T_j(Vx) \cap \{x : V_i \cdot x = V_j \cdot x\}$ for the matrix $V$ and $i, j \in [k]$. We next have to control the Gaussian surface area of the induced shape, i.e., the quantity

$$\Gamma(\{x : j \in \sigma_{1..r}(Vx)\} \cap \{x : V_i \cdot x = V_j \cdot x\})\,.$$

To this end, we give the next lemma.

**Lemma 18.** *Let $r \leq k$ with $r, k \in \mathbb{N}$. For any matrix $\boldsymbol{V} \in \mathbb{R}^{k \times d}$ and $i, j \in [k]$, there exists a matrix $\boldsymbol{Q} = \boldsymbol{Q}^{(i)} \in \mathbb{R}^{k \times d}$ which depends only on $i$ such that*

$$\Gamma(F_i^j) := \Gamma(\{\boldsymbol{x} : j \in \sigma_{1..r}(\boldsymbol{V}\boldsymbol{x})\} \cap \{\boldsymbol{x} : \boldsymbol{V}_i \cdot \boldsymbol{x} = \boldsymbol{V}_j \cdot \boldsymbol{x}\}) \leq 2 \cdot \Pr_{\boldsymbol{x} \sim \mathcal{N}_d}[j \in \sigma_{1..r}(\boldsymbol{Q}\boldsymbol{x})].$$

Before proving this result, let us see how to apply it in order to get Lemma 17. We will have that

$$
\begin{aligned}
\sum_{i \in [k]} \sum_{j \in [k]} \Gamma(F_i^j) &= \sum_{i \in [k]} \sum_{j \in [k]} \Gamma(\{\boldsymbol{x} : j \in \sigma_{1..r}(\boldsymbol{V}\boldsymbol{x})\} \cap \{\boldsymbol{x} : \boldsymbol{V}_i \cdot \boldsymbol{x} = \boldsymbol{V}_j \cdot \boldsymbol{x}\}) \\
&\leq 2 \sum_{i \in [k]} \sum_{j \in [k]} \Pr_{\boldsymbol{x} \sim \mathcal{N}_d}[j \in \sigma_{1..r}(\boldsymbol{Q}^{(i)}\boldsymbol{x})] \\
&= 2 \sum_{i \in [k]} \mathbb{E}_{\boldsymbol{x} \sim \mathcal{N}_d}[|\sigma_{1..r}(\boldsymbol{Q}^{(i)}\boldsymbol{x})|] \\
&= 2 \sum_{i \in [k]} r \\
&= 2kr \, .
\end{aligned}
$$

*Proof of Lemma 18.* For this proof, we fix $i, j \in [k]$. The first step is to design the matrix $\boldsymbol{Q}$. As a first observation, we can subtract the vector $\boldsymbol{V}_i$ from each weight vector and do not affect the resulting orderings. Second, we can assume that the weight vectors that correspond to indices which $j$ beats are unit. Let us be more specific Assume that initially we have that

$$(\boldsymbol{V}_j - \boldsymbol{V}_\ell) \cdot \boldsymbol{x} \geq 0 \, .$$

The first observation gives that

$$(\boldsymbol{V}_j - \boldsymbol{V}_i) \cdot \boldsymbol{x} \geq (\boldsymbol{V}_\ell - \boldsymbol{V}_i) \cdot \boldsymbol{x} \, .$$

Let us set $\widetilde{\boldsymbol{Q}}$ the intermediate matrix with rows $\boldsymbol{V}_j - \boldsymbol{V}_i$. The second observation states that the inequalities where $j$ beats some index $\ell$ are not affected by normalization. Note that $\widetilde{\boldsymbol{Q}}_j \cdot \boldsymbol{x} = 0$ and hence $\widetilde{\boldsymbol{Q}}_\ell \cdot \boldsymbol{x} \leq 0$. Hence, dividing with non-negative numbers will not affect the order of these two values, i.e.,

$$\frac{\widetilde{\boldsymbol{Q}}_j \cdot \boldsymbol{x}}{\|\widetilde{\boldsymbol{Q}}_j\|_2} \geq \frac{\widetilde{\boldsymbol{Q}}_\ell \cdot \boldsymbol{x}}{\|\widetilde{\boldsymbol{Q}}_\ell\|_2} \, .$$

Note that the above ordering is $\boldsymbol{x}$-dependent, since the indices that $j$ beats depend on $\boldsymbol{x}$. However, we can normalize any row of $\widetilde{\boldsymbol{Q}}$ without affecting the fact that the element $j$ is top-$r$ (since the sign of the inner products is not affected by normalization). This transformation yields a matrix $\boldsymbol{Q} = \boldsymbol{Q}^{(i)}$ and depends only on $i$ (crucially, it is independent of $j$). For simplicity, we will omit the index $i$ in what follows. For this matrix, we have that

$$\{\boldsymbol{x} : j \in \sigma_{1..r}(\boldsymbol{Q}\boldsymbol{x}), \boldsymbol{Q}_j \cdot \boldsymbol{x} = 0\} = \{\boldsymbol{x} : j \in \sigma_{1..r}(\boldsymbol{V}\boldsymbol{x}), \boldsymbol{V}_i \cdot \boldsymbol{x} = \boldsymbol{V}_j \cdot \boldsymbol{x}\} \, .$$

We will now prove that

$$\Pr_{\boldsymbol{x} \sim \mathcal{N}_d}[j \in \sigma_{1..r}(\boldsymbol{Q}\boldsymbol{x})] \geq \frac{\Gamma(F_i^j)}{2} \, .$$

Let us fix some $\boldsymbol{x}$ and set $x^\| = \mathrm{proj}_{\boldsymbol{Q}_j} \boldsymbol{x}$ and $x^\perp = \mathrm{proj}_{\boldsymbol{Q}_j^\perp} \boldsymbol{x}$. We assume that $\boldsymbol{x}$ lies in the set $\{\boldsymbol{x} : j \in \sigma_{1..r}(\boldsymbol{Q}\boldsymbol{x})\}$. This implies that there exist an index set $I$ of size at least $k - r$ so that if $\ell \in I$ then

$$\boldsymbol{Q}_j \cdot \boldsymbol{x}^\| + \boldsymbol{Q}_j \cdot \boldsymbol{x}^\perp \geq \boldsymbol{Q}_\ell \cdot \boldsymbol{x}^\| + \boldsymbol{Q}_\ell \cdot \boldsymbol{x}^\perp \, .$$

Let us condition on the event

$$\boldsymbol{Q}_j \cdot \boldsymbol{x}^\perp \geq \boldsymbol{Q}_\ell \cdot \boldsymbol{x}^\perp \, .$$

We hence get that

$$\boldsymbol{Q}_j \cdot \boldsymbol{x}^\| = (\boldsymbol{Q}_j \cdot \boldsymbol{Q}_j) \cdot (\boldsymbol{Q}_j \cdot \boldsymbol{x}) \geq \boldsymbol{Q}_\ell \cdot \boldsymbol{x}^\| = (\boldsymbol{Q}_\ell \cdot \boldsymbol{Q}_j) \cdot (\boldsymbol{Q}_j \cdot \boldsymbol{x})$$

Using that $\boldsymbol{Q}_j$ is unit, that the inner product between $\boldsymbol{Q}_\ell$ and $\boldsymbol{Q}_j$ is at most one and that $\boldsymbol{Q}_j \cdot \boldsymbol{x}$ is a univariate Gaussian, we get that

$$\Pr_{z \sim \mathcal{N}(0,1)}[z \cdot (1 - \boldsymbol{Q}_\ell \cdot \boldsymbol{Q}_j) \geq 0] = 1/2 \,.$$

The above discussion implies that

$$\Pr_{\boldsymbol{x} \sim \mathcal{N}_d}[j \in \sigma_{1..r}(\boldsymbol{Q}\boldsymbol{x})] = \Pr_{\boldsymbol{x} \sim \mathcal{N}_d}[(\forall \ell \in I)\, \boldsymbol{Q}_j \cdot \boldsymbol{x}^\| + \boldsymbol{Q}_j \cdot \boldsymbol{x}^\perp \geq \boldsymbol{Q}_\ell \cdot \boldsymbol{x}^\| + \boldsymbol{Q}_\ell \cdot \boldsymbol{x}^\perp]$$

and so $\Pr_{\boldsymbol{x} \sim \mathcal{N}_d}[j \in \sigma_{1..r}(\boldsymbol{Q}\boldsymbol{x})]$ equals to

$$\Pr_{\boldsymbol{x} \sim \mathcal{N}_d}[(\forall \ell \in I)\, \boldsymbol{Q}_j \cdot \boldsymbol{x}^\| \geq \boldsymbol{Q}_j \cdot \boldsymbol{x}^\| \mid (\forall \ell \in I)\, \boldsymbol{Q}_j \cdot \boldsymbol{x}^\perp \geq \boldsymbol{Q}_\ell \cdot \boldsymbol{x}^\perp] \cdot \Pr_{\boldsymbol{x} \sim \mathcal{N}_d}[(\forall \ell \in I)\, \boldsymbol{Q}_j \cdot \boldsymbol{x}^\perp \geq \boldsymbol{Q}_\ell \cdot \boldsymbol{x}^\perp] \,.$$

However, in the above product, we have that the first term is $1/2$ and the second term is the probability that $j \in \sigma_{1..r}(\boldsymbol{Q}\boldsymbol{x}^\perp)$, i.e.,

$$\Pr_{\boldsymbol{x} \sim \mathcal{N}_d}[j \in \sigma_{1..r}(\boldsymbol{Q}\boldsymbol{x})] \geq \frac{\Pr[j \in \sigma_{1..r}(\boldsymbol{Q}\boldsymbol{x}^\perp)]}{2} = \Gamma(F_i^j)/2 \,,$$

since the space in the RHS is low-dimensional and corresponds to the desired surface. $\qquad\square$

# D   Distribution-Free Lower Bounds for Top-1 Disagreement Error

We begin with some definitions concerning the PAC Label Ranking setting. Let $\mathcal{X}$ be an instance space and $\mathcal{Y} = \mathbb{S}_k$ be the space of labels, which are rankings over $k$ elements. A sorting function or hypothesis is a mapping $h : \mathcal{X} \to \mathbb{S}_k$. We denote by $h_1(x)$ the top-1 element of the ranking $h(x)$. A hypothesis class is a set of classifiers $\mathcal{H} \subset \mathbb{S}_k^{\mathcal{X}}$.

**Top-1 Disagreement Error.** The top-1 disagreement error with respect to a joint distribution $\mathcal{D}$ over $\mathcal{X} \times \mathbb{S}_k$ equals to the probability $\Pr_{(x,\sigma) \sim \mathcal{D}}[h_1(x) \neq \sigma^{-1}(1)]$. We mainly consider learning in the **realizable** case, which means that there is $h^\star \in \mathcal{H}$ which has (almost surely) zero error. Therefore, we can focus on the marginal distribution $\mathcal{D}_x$ over $\mathcal{X}$ and denote the top-1 disagreement error of a sorting function $h$ with respect to the true hypothesis $h^\star$ by $\mathrm{Err}_{\mathcal{D}_x, h^\star}(h) := \Pr_{x \sim \mathcal{D}_x}[h_1(x) \neq h_1^\star(x)]$.

A learning algorithm is a function $\mathcal{A}$ that receives a training set of $m$ instances, $S \in \mathcal{X}^m$, together with their labels according to $h^\star$. We denote the restriction of $h^\star$ to the instances in $S$ by $h^\star|_S$. The output of the algorithm $\mathcal{A}$, denoted $\mathcal{A}(S, h^\star|_S)$ is a sorting function. A learning algorithm is proper if it always outputs a hypothesis from $\mathcal{H}$.

The top-1 PAC Label Ranking sample complexity of a learning algorithm $\mathcal{A}$ is the function $m_{\mathcal{A},\mathcal{H}}^{(1)}$ defined as follows: for every $\epsilon, \delta > 0$, $m_{\mathcal{A},\mathcal{H}}^{(1)}(\epsilon, \delta)$ is the minimal integer such that for every $m \geq m_{\mathcal{A},\mathcal{H}}^{(1)}(\epsilon, \delta)$, every distribution $\mathcal{D}_x$ on $\mathcal{X}$, and every target hypothesis $h^\star \in \mathcal{H}$, $\Pr_{S \sim \mathcal{D}_x^m}[\mathrm{Err}_{\mathcal{D}_x, h^\star}(\mathcal{A}(S, h^\star|_S)) > \epsilon] \leq \delta$. In this case, we say that the learning algorithm $(\epsilon, \delta)$-learns the class of sorting functions $\mathcal{H}$ with respect to the top-1 disagreement error. If no integer satisfies the inequality above, define $m_{\mathcal{A}}^{(1)}(\epsilon, \delta) = \infty$. $\mathcal{H}$ is learnable with $\mathcal{A}$ if for all $\epsilon$ and $\delta$ the sample complexity is finite. The **top-1 PAC Label Ranking sample complexity** of a class $\mathcal{H}$ is $m_{\mathrm{PAC},\mathcal{H}}^{(1)}(\epsilon, \delta) = \inf_{\mathcal{A}} m_{\mathcal{A},\mathcal{H}}^{(1)}(\epsilon, \delta)$, where the infimum is taken over all learning algorithms. Clearly, the above top-1 definition can be extended to the top-$r$ setting.

In this section, we show the next result. We denote by $\mathcal{L}_{d,k}$ the class of Linear Sorting functions in $d$ dimensions with $k$ labels.

**Theorem 7.** *In the realizable PAC Label Ranking setting, any algorithm that $(\epsilon, \delta)$-learns the class $\mathcal{L}_{d,k}$ with respect to the top-1 disagreement error requires at least $\Omega((dk + \log(1/\delta))/\epsilon)$ samples.*

## D.1   Top-1 Ranking Natarajan Dimension

In order to establish the above result, we introduce a variant of the standard Natarajan dimension [Nat89, BDCBL92, DSBDSS11, DSS14]. For a ranking $\pi$, we will also let $L_1(\pi)$ its top-1 element and $L_{3..k}(\pi)$ the ranking after deleting its top-2 part.

**Definition 3** (Top-1 Ranking Natarajan Dimension). *Let $\mathcal{H} \subseteq \mathbb{S}_k^{\mathcal{X}}$ be a hypothesis class of sorting functions and let $S \subseteq \mathcal{X}$. We say that $\mathcal{H}$ N-shatters $S$ if there exist two mappings $f_1, f_2 : S \to \mathbb{S}_k$ such that for every $y \in S$, $L_1(f_1(y)) \neq L_1(f_2(y))$ and $L_{3..k}(f_1(y)) = L_{3..k}(f_2(y))$ and for every $T \subseteq S$, there exists a sorting function $g \in \mathcal{H}$ such that*

$$(i) \; \forall x \in T, \;\; g(x) = f_1(x), \text{ and } (ii) \; \forall x \in S \setminus T, \;\; g(x) = f_2(x) \,.$$

*The **top-1 Ranking Natarajan dimension** of $\mathcal{H}$, denoted $d_N^{(1)}(\mathcal{H})$ is the maximal cardinality of a set that is N-shattered by $\mathcal{H}$.*

First, we connect PAC Label Ranking learnability to the top-1 disagreement error with the notion of top-1 Ranking Natarajan dimension.

**Theorem 8** (Top-1-Natarajan Lower Bounds Sample Complexity). *In the realizable PAC Label Ranking setting, we have for every hypothesis class $\mathcal{H} \subseteq \mathbb{S}_k^{\mathcal{X}}$*

$$m_{\mathrm{PAC},\mathcal{H}}^{(1)}(\epsilon, \delta) = \Omega\left(\frac{d_N^{(1)}(\mathcal{H}) + \ln(1/\delta)}{\epsilon}\right) \,.$$

*Proof.* Let $\mathcal{H} \subseteq \mathbb{S}_k^{\mathcal{X}}$ be a class of sorting functions of top-1-Natarajan dimension $d_N^{(1)} = d_N$. Consider the binary hypothesis class $\mathcal{H}_{\mathrm{bin}} = \{0,1\}^{[d_N]}$ which contains all the classifiers from $[d_N] = \{1, ..., d_N\}$ to $\{0,1\}$. It suffices to show the following.

**Claim 12.** *It holds that $m_{\mathrm{PAC},\mathcal{H}}^{(1)}(\epsilon, \delta) \geq m_{\mathrm{PAC},\mathcal{H}_{\mathrm{bin}}}(\epsilon, \delta)$.*

This is sufficient since we have that $m_{\mathrm{PAC},\mathcal{H}_{\mathrm{bin}}}(\epsilon, \delta) = \Omega\left(\frac{\mathrm{VC}(\mathcal{H}_{\mathrm{bin}}) + \ln(1/\delta)}{\epsilon}\right)$ and $\mathrm{VC}(\mathcal{H}_{\mathrm{bin}}) = d_N$. Let us now prove the claim.

We assume that the instance space is the set $\mathcal{X}$. Assume that $A$ is a learning algorithm for the hypothesis class $\mathcal{H} \subseteq \mathbb{S}_k^{\mathcal{X}}$ and $A_{\mathrm{bin}}$ is a learning algorithm for the associated binary class $\mathcal{H}_{\mathrm{bin}}$. It suffices to show that $A$ requires at least as many samples as $A_{\mathrm{bin}}$. In fact, we will show that whenever $A_{\mathrm{bin}}$ errs, so does $A$. Let $S = \{s_1, ..., s_{d_N}\}$, $f_0, f_1$ be the set and the two functions that witness that the top-1-Natarajan dimension of $\mathcal{H}$ is $d_N$. Given a training set $(x_i, y_i)_{i \in [m]} \in ([d_N] \times \{0,1\})^m$, we set $g : \mathcal{X} \to \mathbb{S}_k$ be equal to the output of the algorithm $A$ with input $(s_{x_i}, f_{y_i}(x_i))_{i \in [m]} \in (S \times \mathbb{S}_k)^m$. We also set $f$ be the output of the algorithm $A_{\mathrm{bin}}$ with input $(x_i, y_i)_{i \in [m]}$ by setting $f(i) = 1$ if and only if $L_1(g(s_i)) = L_1(f_1(s_i))$. We will show that whenever $A_{\mathrm{bin}}$ errs, so does $A$. Fix $(x_i, y_i) \in S \times \{0,1\}$. Assume that $A_{\mathrm{bin}}(x_i) \neq y_i$ and say $y_i = 0$. Then $f(i) = 1$ and so $L_1(g(s_i)) = L_1(f_1(s_i)) \neq L_1(f_0(s_i))$. This implies that $A$ errs. The case $y_i = 1$ is similar. $\square$

### D.2 Lower Bound for top-1 disagreement error for LSFs

**Theorem 9** (Top-1 Natarajan Dimension of LSFs). *Consider the hypothesis class $\mathcal{L}_{d,k} = \{\sigma_{\boldsymbol{W}} : \mathbb{R}^d \to \mathbb{S}_k : \sigma_{\boldsymbol{W}}(\boldsymbol{x}) = \mathrm{argsort}(\boldsymbol{W}\boldsymbol{x}), \boldsymbol{W} \in \mathbb{R}^{k \times d}\}$. Then, $d_N^{(1)}(\mathcal{L}_{d,k}) = \Omega(dk)$.*

*Proof.* Fix $k \in \mathbb{N}$. Let us consider the case $d = 2$ that will correspond as the building block for the general case $d > 2$. Let us first choose the set of points: Set $P$ be the collection of pairs $P = \{(2i-1, 2i)\}_{i \in [b]}$ for any $i \in [b]$ with $b = \lfloor k/2 \rfloor$ and $S = \{\boldsymbol{x}_m\}_{m \in P}$ where these points correspond to $|P|$ equidistributed points on the unit sphere in $\mathbb{R}^2$. This set of points has size $|P| = \Theta(k)$ and we are going to N-shatter it using $\mathcal{L}_{2,k}$.

Consider the matrix $\boldsymbol{W} \in \mathbb{R}^{k \times 2}$ so that $\{\boldsymbol{W}_i\}_{i \in [k]}$ correspond to the rows of $\boldsymbol{W}$. The structure of the problem relies on the hyperplanes with normal vectors $(\boldsymbol{W}_i - \boldsymbol{W}_j)_{i \neq j}$ and our choice of $\boldsymbol{W}$ will rely on these hyperplanes. For any $m = (2i-1, 2i)$, we set $\boldsymbol{W}_{2i-1}, \boldsymbol{W}_{2i}$ on the unit sphere so that $\boldsymbol{W}_{2i-1} \cdot \boldsymbol{W}_{2i} = 1 - \phi$ with $\phi \in (0,1)$ sufficiently small (set $\arccos(1 - \phi) = 2\pi/(100k)$) and let $C_m$ be the cone generated by these two vectors with axis $I_m$. We place $\boldsymbol{W}_{2i-1}$ so that the distance between $\boldsymbol{x}_m$ and the hyperplane $I_m$ is sufficiently small (say that the angle between $\boldsymbol{x}_m$ and $I_m$ is $\arccos(1 - \phi)/100$). Note that the normal vector of $I_m$ is $\boldsymbol{W}_{2i-1} - \boldsymbol{W}_{2i}$ and we place $\boldsymbol{x}_m$ so that it has positive correlation with this vector. This uniquely identifies the location of $\boldsymbol{W}_{2i}$. Crucially, each vector $\boldsymbol{x}_m$ has the following properties: (i) $\boldsymbol{x}_m$ is very close to the boundary of the hyperplane

with normal vector $(\boldsymbol{W}_{2i-1} - \boldsymbol{W}_{2i})$, (ii) $\boldsymbol{W}_{2i-1} \cdot \boldsymbol{x}_m > \boldsymbol{W}_{2i} \cdot \boldsymbol{x} > \boldsymbol{W}_j \cdot \boldsymbol{x}_m$ for any $j \notin m$ and (iii) $\boldsymbol{x}_m$ is far from any boundary induced by hyperplanes with normal vectors $\boldsymbol{W}_j - \boldsymbol{W}_{j'}$ for any $(j, j') \neq m$.

Since the points are well-separated on the unit sphere, for any $m = (2i - 1, 2i) \in P$, we have $\boldsymbol{W}_{2i-1} \cdot \boldsymbol{W}_{2i} = 1 - \phi \approx 1$ and for any other pair of indices $(i, j) \notin P$, there exists $c = c(k) \in (0, 1)$, $|\langle \boldsymbol{W}_i, \boldsymbol{W}_j \rangle| \leq c$.

For any $m = (2i - 1, 2i) \in P$, we set $\boldsymbol{W}'_{2i-1} - \boldsymbol{W}'_{2i} = \boldsymbol{R}_\theta(\boldsymbol{W}_{2i-1} - \boldsymbol{W}_{2i})$ for some $\theta$ to be chosen, where $\boldsymbol{R}_\theta$ is the $2 \times 2$ rotation matrix. We choose $\theta$ so that each point $\boldsymbol{x}_m$ for $m = (2i - 1, 2i) \in P$ with $(\boldsymbol{W}_{2i-1} - \boldsymbol{W}_{2i}) \cdot \boldsymbol{x}_m > 0$ satisfies $(\boldsymbol{W}'_{2i-1} - \boldsymbol{W}'_{2i}) \cdot \boldsymbol{x}_m < 0$. The main idea is that since $\boldsymbol{x}_m$ has the properties (i)-(iii) described above, the rankings induced by the vectors $\boldsymbol{W}\boldsymbol{x}_m$ and $\boldsymbol{W}'\boldsymbol{x}_m$ will be different in the first two positions but the same in the rest.

Given the training set $\{\boldsymbol{x}_m\}_{m \in P}$, we have to construct $f_0, f_1$ and verify that they satisfy the top-1 Ranking Natarajan conditions. For $m = (2i - 1, 2i)$, we have that $f_0(\boldsymbol{x}_m) = (2i - 1, 2i, \pi)$ and $f_1(\boldsymbol{x}_m) = (2i, 2i - 1, \pi)$ for some ranking $\pi$ of size $k - 2$ that depends on $m$. Specifically, we will set $f_0(\boldsymbol{x}) = \sigma(\boldsymbol{W}\boldsymbol{x})$ and $f_1(\boldsymbol{x}) = \sigma(\boldsymbol{W}'\boldsymbol{x})$, where $\sigma$ gives the decreasing ordering of the elements of the input vector. By the choice of the set $S$ and $\boldsymbol{W}, \boldsymbol{W}'$, it remains to show that the $k - 2$ last elements of the rankings $f_0(\boldsymbol{x}_m)$ (say $\pi_0$) and of $f_1(\boldsymbol{x}_m)$ (say $\pi_1$) are in the same order, i.e., $L_{3..k}(f_0(\boldsymbol{x}_m)) = L_{3..k}(f_1(\boldsymbol{x}_m))$. Assume that $u \succ v$ in $\pi_0$. It suffices to show that $(\boldsymbol{W}'_u - \boldsymbol{W}'_v) \cdot \boldsymbol{x}_m \geq 0$, i.e., the order of $u$ and $v$ is preserved when transforming $\boldsymbol{W}$ to $\boldsymbol{W}'$. We have that $(\boldsymbol{W}_u - \boldsymbol{W}_v) \cdot \boldsymbol{x}_m > c_1$ for some constant $c_1 > 0$ ($c_1$ is the minimum over $(u, v) \neq m = (2i - 1, 2i)$). Hence, we can pick $\theta$ small enough so that $(\boldsymbol{W}'_u - \boldsymbol{W}'_v) \cdot \boldsymbol{x}_m > c_2$ and this can be done for any pair $u, v$ that does not correspond to $m$. This implies that $\pi_0 = \pi_1 = \pi$. In particular, we have that

$$(\boldsymbol{W}'_u - \boldsymbol{W}'_v) \cdot \boldsymbol{x}_m = \cos(\theta) \cdot (\boldsymbol{W}_u - \boldsymbol{W}_v) \cdot \boldsymbol{x}_m + \sin(\theta) \cdot (W_{uv}^{(1)} x_m^{(2)} - W_{uv}^{(2)} x_m^{(1)}) > c_2 > 0$$

for some $\theta$ sufficiently small, where $W_{uv}^{(t)}$ is the $t$-th entry of the vector $\boldsymbol{W}_u - \boldsymbol{W}_v$ for $t \in \{1, 2\}$ and $\boldsymbol{x}_m, \boldsymbol{W}_u, \boldsymbol{W}_v$ are unit vectors.

For any subset $T$ of $S$, it remains to choose a linear classifier in $\mathcal{L}_{2,k}$ (which is allowed to depend on $T$). For any $T \subseteq S = \{\boldsymbol{x}_m\}_{m \in P}$, we consider the matrix $\overline{\boldsymbol{W}} \in \mathbb{R}^{k \times 2}$ so that for the $i$-th row $\overline{W}_i = \boldsymbol{W}_i \mathbb{1}\{i \in m \in T\} + \boldsymbol{W}'_i \mathbb{1}\{i \in m \in S \setminus T\}$ for any $i \in [k]$. This is valid since the pairs $m \in P$ partition $[k]$. We have to show the following two properties: (i) $\sigma(\overline{\boldsymbol{W}}\boldsymbol{x}) = f_0(\boldsymbol{x})$ for $x \in T$ and (ii) $\sigma(\overline{\boldsymbol{W}}\boldsymbol{x}) = f_1(\boldsymbol{x})$ for $x \in S \setminus T$.

Assume that $m = (2i - 1, 2i)$ and $\boldsymbol{x}_m \in T$. We have that $f_0(\boldsymbol{x}_m) = (2i - 1, 2i, \pi)$ and $\overline{\boldsymbol{W}}_{2i-1} - \overline{\boldsymbol{W}}_{2i} = \boldsymbol{W}_{2i-1} - \boldsymbol{W}_{2i}$ and so $2i - 1 \succ 2i$ in the ranking $\sigma(\overline{\boldsymbol{W}}\boldsymbol{x}_m)$. It remains to show that the remaining $\binom{k}{2} - 1$ pairwise comparisons are the same in the two rankings. Let us consider a pair of points $u \neq v$ so that $u \succ v$ in $f_0(\boldsymbol{x}_m)$. It suffices to show that $u \succ v$ in $\sigma(\overline{\boldsymbol{W}}\boldsymbol{x}_m)$.

1. If $u, v$ are so that $\overline{\boldsymbol{W}}_u - \overline{\boldsymbol{W}}_v = \boldsymbol{W}_u - \boldsymbol{W}_v$, the result holds.

2. If $u, v$ are so that $\overline{\boldsymbol{W}}_u - \overline{\boldsymbol{W}}_v = \boldsymbol{W}_u - \boldsymbol{W}'_v$: In this case, $u$ and $v$ lie in a different pair of $P$ and this implies that the correct direction is preserved if $\theta$ is appropriately chosen. For $\theta$ as above, it holds that $(\boldsymbol{W}_u - \boldsymbol{R}_\theta \boldsymbol{W}_v) \cdot \boldsymbol{x}_m$ has the same sign as $(\boldsymbol{W}_u - \boldsymbol{W}_v) \cdot \boldsymbol{x}_m$. In particular,

$$\boldsymbol{W}_u \cdot \boldsymbol{x}_m - \boldsymbol{R}_\theta \boldsymbol{W}_v \cdot \boldsymbol{x}_m = \boldsymbol{W}_u \cdot \boldsymbol{x}_m - (\cos(\theta) W_v^{(1)} - \sin(\theta) W_v^{(2)}) x_m^{(1)} - (\sin(\theta) W_v^{(1)} + \cos(\theta) W_v^{(2)}) x_m^{(2)},$$

and so

$$(\boldsymbol{W}_u - \boldsymbol{W}'_v) \cdot \boldsymbol{x}_m = \cos(\theta) \cdot (\boldsymbol{W}_u - \boldsymbol{W}_v) \cdot \boldsymbol{x}_m + \sin(\theta)(W_v^{(2)} x_m^{(1)} - W_v^{(1)} x_m^{(2)}) > 0.$$

3. If $u, v$ are so that $\overline{\boldsymbol{W}}_u - \overline{\boldsymbol{W}}_v = \boldsymbol{W}'_u - \boldsymbol{W}'_v$, the analysis for the inner product with $\boldsymbol{x}_m$ will be similar.

We now have to extend this proof for $d > 2$. We will "tensorize" the above construction as follows. Let $S = \{\boldsymbol{y}_{mj}\}_{m \in [b], j \in [d/2]}$ with $|S| = \lfloor k/2 \rfloor \cdot \lfloor d/2 \rfloor$. We first define the points of $S$: For $s \in [d]$,

set $y_{mj}[s] = x_m[1]\mathbb{1}\{s = 2j - 1\} + x_m[2]\mathbb{1}\{s = 2j\}$ with $\boldsymbol{y}_{mj} \in \mathbb{R}^d$, i.e., $\boldsymbol{y}_{mj}$ has the values of $\boldsymbol{x}_m$ at the consecutive entries indicated by $m = (2i - 1, 2i) \in P$ and zeros at the other positions.

We have to show that the set $S$ is $N$-shattered. Given $T \subseteq S$, we are going to create the matrix $\overline{\boldsymbol{W}} \in \mathbb{R}^{k \times d}$. For illustration, think of each row of the matrix as having $d/2$ blocks of size two. If $\boldsymbol{y}_{mj} \in T$ with $m = (2i-1, 2i)$, set the two associated rows (indicated by $m$) of $\overline{\boldsymbol{W}}$ with $\boldsymbol{W}_{2i-1}, \boldsymbol{W}_{2i}$ at the $j$-th block and with $\boldsymbol{W}'_{2i-1}, \boldsymbol{W}'_{2i}$ otherwise. We will have that $\sigma(\overline{\boldsymbol{W}}\boldsymbol{y}) = f_0(\boldsymbol{y})$ if $\boldsymbol{y} \in T$ and $\sigma(\overline{\boldsymbol{W}}\boldsymbol{y}) = f_1(\boldsymbol{y})$ otherwise and the analysis is the same as the $d = 2$ case. $\qquad \square$

# E  Examples of Noisy Ranking Distributions

**Definition 4** (Mallows model [Mal57]). *Consider $k$ alternatives and let $\pi \in \mathbb{S}_k, \phi \in [0, 1]$. The Mallows distribution $\mathcal{M}_{\mathrm{Mal}}(\pi, \phi)$ with central ranking $\pi$ and spread parameter $\phi$ is a probability measure over $\mathbb{S}_k$ with density $\mathbf{Pr}_{\sigma \sim \mathcal{M}_{\mathrm{Mal}}(\pi, \phi)}[\sigma]$ that is proportional to $\phi^{d(\sigma, \pi)}$, where $d$ is a ranking distance.*

We focus on Mallows models accociated with the Kendall's Tau distance $d = d_{KT}$ (the standard distance, not the normalized one), which measures the number of discordant pairs.

**Fact 2.** *When $\phi < 1$, the Mallows model $\mathcal{M}_{\mathrm{Mal}}(\pi, \phi)$ is a ranking distribution with bounded noise at most $\frac{1+\phi}{4} < 1/2$.*

*Proof.* The following property holds [Mal57]

$$\mathbf{Pr}_{\sigma \sim \mathcal{M}_{\mathrm{Mal}}(\pi, \phi)}[\sigma(i) < \sigma(j)|\pi(i) < \pi(j)] = \frac{\pi(j) - \pi(i) + 1}{1 - \phi^{\pi(j) - \pi(i) + 1}} - \frac{\pi(j) - \pi(i)}{1 - \phi^{\pi(j) - \pi(i)}} \geq \frac{1}{2} + \frac{1 - \phi}{4}.$$

$\qquad \square$

The Bradley-Terry-Luce model [BT52, Luc12] is the most studied pairwise comparisons model. In his seminal paper, Mallows [Mal57] also studied the following natural ranking distribution:

**Definition 5** (Bradley-Terry-Mallows [Mal57]). *Consider a score vector $\boldsymbol{w} \in \mathbb{R}_+^k$ with $k$ distinct entries and let $\pi$ be the ranking induced by the values of $\boldsymbol{w}$ in decreasing order. The Bradley-Terry-Mallows distribution $\mathcal{M}_{\mathrm{BTM}}(\boldsymbol{w})$ with central ranking $\pi$ is a probability measure over $\mathbb{S}_k$ with density $\mathbf{Pr}_{\sigma \sim \mathcal{M}_{\mathrm{BTM}}(\boldsymbol{w})}[\sigma]$ that is proportional to $\prod_{i \succ_\sigma j} \frac{w_i}{w_i + w_j}$.*

**Lemma 19.** *There exists a real number $0 < \eta < 1/2$ so that the Bradley-Terry-Mallows distribution $\mathcal{M}_{\mathrm{BTM}}(\boldsymbol{w})$ is a ranking distribution with bounded noise at most $\eta$.*

*Proof.* In the standard Bradley-Terry-Luce model, the pairwise comparison between the alternatives $i, j$ is a Bernoulli random variable with $\mathbf{Pr}[i \succ j] = w_i/(w_i + w_j)$. The Bradley-Terry-Mallows distribution can be considered as the Bradley-Terry-Luce model conditioned on the event that all the pairwise comparisons are consistent to a ranking. Hence, we have that

$$\mathbf{Pr}_{\sigma \sim \mathcal{M}_{\mathrm{BTM}}(\boldsymbol{w})}[\sigma] = \frac{1}{Z(k, \boldsymbol{w})} \prod_{i \succ_\sigma j} \frac{w_i}{w_i + w_j}.$$

Let us set $\mathcal{A}_{i \succ j} = \{\sigma \in \mathbb{S}_k : \sigma(i) < \sigma(j)\}$. We are interested in the following probability

$$\mathbf{Pr}_{\sigma \sim \mathcal{M}_{\mathrm{BTM}}(\boldsymbol{w})}[i \succ_\sigma j|w_i > w_j] = \mathbf{Pr}_{\sigma \sim \mathcal{M}_{\mathrm{BTM}}(\boldsymbol{w})}[\sigma(i) < \sigma(j)|w_i > w_j] = \frac{1}{Z(k, \boldsymbol{w})} \sum_{\sigma \in \mathcal{A}_{i \succ j}} \prod_{p \succ_\sigma q} \frac{w_p}{w_p + w_q}.$$

Note that in order to show the desired property, it suffices to show that

$$\sum_{\sigma \in \mathcal{A}_{i \succ j}} \prod_{p \succ_\sigma q} \frac{w_p}{w_p + w_q} > \sum_{\sigma \in \mathcal{A}_{i \prec j}} \prod_{p \succ_\sigma q} \frac{w_p}{w_p + w_q}.$$

First, observe that there exists a correspondence mapping $\sigma \in \mathcal{A}_{i \succ j}$ to $\mathcal{A}_{i \prec j}$, where one flips the elements $i$ and $j$. Hence, it suffices to show that the mass of the ranking $(u_a)i(u_b)j(u_c)$ is larger than the one of the ranking $(u_a)j(u_b)i(u_c)$, where $u_a, u_b, u_c$ are permutations of length between 0 and

$k - 2$ with elements in $[k] \setminus \{i, j\}$. For the two above rankings, the only terms of the product that are not identical are the following

$$\frac{w_i}{w_i + w_j} \prod_{x \in u_b} \frac{w_i}{w_i + w_x} \frac{w_x}{w_x + w_j} > \frac{w_j}{w_i + w_j} \prod_{x \in u_b} \frac{w_j}{w_j + w_x} \frac{w_x}{w_x + w_i},$$

since $w_i > w_j$ and so the result follows. □