# OpenReview forum: "Linear Label Ranking with Bounded Noise"
_NeurIPS.cc/2022/Conference — NeurIPS 2022 Accept_

### Official Review · Reviewer_pDth · 2022-07-11

**Rating:** 7
**Confidence:** 3
**Soundness:** 4 excellent
**Presentation:** 3 good
**Contribution:** 3 good

**Summary:**

The setup is the following: there is an unknown k x d matrix W and a player observes a feature vector x in R^d and a 'ranking' sigma(x) (i.e., a permutation) over [k] generated as follows. The feature vector x is sampled from a d-dimensional standard Gaussian. Then, the permutation sigma(x) over [k] is generated by sorts the indices of Wx in decreasing order.

The goal is the learn a matrix ~W which approximates the label ranking, in particular, we want that with high probability over a fresh x drawn from a d-dimensional standard Gaussian, sorting ~Wx gives a permutation which is very close to that of Wx.

The paper studies two notions of closeness: kendal-tau distance, and top-r distance.

The kendal-tau distance (KT) between two permutations is the fraction of pairs i,j in [k] where their relative order agrees. In the learning setup, this corresponds to saying that with high probability over x, with high probability over a random (i,j) from [k]x[k], the relative order of (Wx)_i and (Wx)_j agrees with (~Wx)_i and (~Wx)_j.

While the kendal-tau distance is well-studied, it is perhaps less motivated in ranking setups, where one is more interested in higher ranked elements. In settings where higher ranked elements are more important, the paper studies the top-r distance. This is a 0-1 distance based on whether the top r ranked elements are exactly the same in exactly the same order.

While exact versions of the above are relatively simple (an algorithm using linear programming can find ~W), there is some noise in what the player observes. In particular, the player observes a draw from a distribution which is promised that each pair disagrees with the ground-truth ordering with probability at most eta (where eta < 1/2).

Results:

1. A polynomial-time algorithm for learning ~W in KT distance from O(d log(k) / (eps (1 - 2eta)^6 ) ) up to distance eps.

2. A polynomial-time algorithm for learning ~W in top-r distance from O(d k r / (eps (1-2eta)^6 ) ) up to distance eps.

Important remarks:

The noise model is arbitrary as long as it has marginals on pairs which are different with probability at most eta. This, along with the fact that sorting functions are linear, makes the problem a similar of learning halfspaces with Massart noise.

Because of this connection, the assumption that x is Gaussian is somewhat necessary because there are  super-polynomial lower bounds in the statistical query model.

The algorithm proceeds in three steps. First, a reduction from a ranking to O(k^2) binary comparisons. Second, an improper learner which aggregates the O(k^2) binary comparisons. Third, an algorithm which uses the intermediate steps of the improper learning to output a hypothesis ~W. While the first and second steps are known and have appeared in the literature before, the novel aspect of this work is finding the matrix ~W -- to do this, the paper proves two interesting geometric lemmas relating the angles between proposed rows of ~W and W with the corresponding KT and top-r distance.


**Questions:**

I don't have any pressing questions.

**Limitations:**

The work is purely theoretical at this point, and seems to have no potential negative societal impact.

**Strengths And Weaknesses:**

Strengths:

The paper studies a natural problem in learning rankings. The problems seem like natural extensions of learning halfspaces with Massart noise, and a good model for learning rankings with noise. From a technical perspective, the approach is natural and the geometric lemmas interesting. The paper is also well-written.

Weaknesses:

I don't really see any strong weaknesses in the paper.

---

> ### Author Response · Authors · 2022-08-02
> **Thank you for your review!**
>
> We would like to thank the reviewer for carefully reading our manuscript and for the useful and positive feedback!

---

### Official Review · Reviewer_A2Wu · 2022-07-11

**Rating:** 8
**Confidence:** 4
**Soundness:** 3 good
**Presentation:** 4 excellent
**Contribution:** 4 excellent

**Summary:**

This paper considers the learning of linear sorting functions under Gaussian marginals in presence of bounded noise. In the special case k=2, the problem reduces to the well-studied learning of halfspaces with Massart noise. The author generalized the problem setting and provided efficient algorithms with respect to Kendall’s Tau distance and top-r disagreement loss.


**Questions:**

The current work studies bounded noise type. Is there any possibility or evidence of extending it to more challenging noise types, e.g. adversarial noise?

Minor issue: Some notations are not well defined. For example, the sample complexity N could be distinct for different algorithms. r was used both for the top-r problem and for the radius in Claim 2.


**Limitations:**

The work does not have negative social impacts.


**Strengths And Weaknesses:**

The work makes a significant contribution by proposing the first efficient algorithm for learning of LSFs with bounded noise. The basic algorithmic ingredient is an efficient learner ([ZSA20]) for the class of halfspaces (for the special case of k=2). However, the algorithm is generalized to any k (improperly), and is further used to obtain a proper learner using the ellipsoid method. When the error is measured by top-r disagreement loss, the proper learner also achieves improved sample complexity comparing to a naive invocation of the improper learner. The paper is very well-written with technical highlights appropriately placed and the analysis is sound.

---

> ### Author Response · Authors · 2022-08-02
> **Adversarial Noise**
>
> We thank the reviewer for carefully reading our manuscript and providing insightful feedback (and also some interesting questions!).
>
> "The current work studies bounded noise type. Is there any possibility or evidence of extending it to more challenging noise types, e.g. adversarial noise?"
>
> That is indeed a very interesting question for direct future research. While in general adversarial noise models have strong lower bounds, in the presence of Gaussian marginals or more generally structured distributions, approximate learnability is known to be achievable for binary classifiers with adversarial label noise.  It is very interesting to investigate whether such approximate learning results can be obtained for multiclass/label ranking settings under structured distributions.  We believe that the techniques developed in this work can serve as a starting point towards this goal.
>
> Minor issue: Some notations are not well defined. For example, the sample complexity N could be distinct for different algorithms. r was used both for the top-r problem and for the radius in Claim 2.
>
> We will fix this (and potential similar issues) in the first revision of our work.

---

### Official Review · Reviewer_bxrG · 2022-07-12

**Rating:** 8
**Confidence:** 3
**Soundness:** 4 excellent
**Presentation:** 4 excellent
**Contribution:** 4 excellent

**Summary:**

This paper is the first to study the problem of learning linear label rankings in the presence of noise.

In the label ranking problem, we are given access to samples of the form $(x,y)$ where $x \in \mathbb{R}^d$ and $y$ is a permutation of the sequence {$1, 2, 3, \ldots, k$}. For example, this can correspond to a ranking of movies by preferences of a particular user in a movie recommendation system. In the **linear** label ranking problem, there is an additional constraint that the ranking should be such that it can be formed by the indices corresponding to a descending sort of the entries of $Wx$ for some matrix $W \in \mathbb{R}^{k \times d}$. Further, in the **noisy** linear label ranking problem, we are given access not to *pure* samples from a linear label ranking distribution but instead samples whose labels are corrupted by some noise. This paper also assumes that the marginal distribution of $x$ needs to be Gaussian.

They provide two algorithms, one improper and one proper, for learning with error bounds in the normalized Kendall tau (KT) distance. They also provide an algorithm with error bounds in the top-$r$ disagreement metric. In particular, their improper learning algorithm in the KT distance uses algorithms for learning linear-threshold-functions (LTFs) in the Massart noise model as sub-routines.

**Questions:**

Not any as of now.

**Limitations:**

This is primarily a theoretical paper and so the authors have mentioned that it doesn't have any negative social impact.

**Strengths And Weaknesses:**

Originality: I am not an expert in this area so I am not entirely sure about other related work. The proposed algorithms and getting them to work (as in proving guarantees for them) are quite non-trivial and so the paper is quite original in my opinion.

Quality: The submission is technically sound. All claims are well-supported with proofs.

Clarity: The submission is clearly written and well-organized.

Significance: The paper is the first to study a very natural problem and so I think it is quite significant. Ranking functions have many applications and developing robust algorithms for learning ranking functions can have good practical impact. On the theoretical front, these problems are also clearly of interest to the NeurIPS community. As mentioned on page 2 of the paper, the case of $k = 2$ captures the problem of learning halfspaces with Massart noise - the best paper award winner of NeurIPS 2019 was on this topic.

---

> ### Author Response · Authors · 2022-08-02
> **Thank you for your review**
>
> We would like to thank the reviewer for carefully reading our manuscript and for appreciating our work and results!

---

### Official Review · Reviewer_AkRj · 2022-07-12

**Rating:** 7
**Confidence:** 2
**Soundness:** 4 excellent
**Presentation:** 3 good
**Contribution:** 3 good

**Summary:**

This work tackles the problem of learning linear sorting functions with bounded noise under Gaussian martingales. The proposed algorithms enjoy strong theoretical sampling guarantees and a polynomial runtime, for both the normalized Kendall’s Tau distance and the top-r disagreement loss.


**Questions:**

- What is the intuition behind the power 6 in theorem 1 and 2?
- Is Algorithm 2 used for both Theorem 1 and 2?


**Limitations:**

The theoretical limitations are adequately addressed. The authors state that the potential negative societal impacts of their work is N/A due to its theoretical nature. It might still be valuable to mention what could go wrong if the suggested algorithms were actually deployed.


**Strengths And Weaknesses:**

Strengths:
- Presentation: the problem is well introduced and the main results are clearly presented
- Impact: the results established seem to be of general interest in addition to solve the label ranking problem
- The paper is technically sound.

Weaknesses:
- No experimental results limit the impact of the work.
- Clarity: although the first two sections are very clear, the second half of the paper feels harder to follow. It does not feel clear to me whether the stated algorithms are solutions to the problem with KT Distance or with top-r Disagreement, or both.

---

> ### Author Response · Authors · 2022-08-02
> **Experiments, Clarification on Theorems 1, 2, Intuition behind the sample complexity**
>
> We thank the reviewer for recognizing the importance of the noisy label ranking problem and the fact that the work is clearly presented and written. We would like to respond to the reviewer's questions and comments inline, as follows:
>
> "No experimental results limit the impact of the work."
>
> The main goal of this work is to provide the first efficient algorithms for learning noisy LSFs with provable guarantees.  We agree with the reviewer that experimental evaluation is interesting and we plan to further investigate and implement our algorithms in a future work.
>
> "Clarity: although the first two sections are very clear, the second half of the paper feels harder to follow. It does not feel clear to me whether the stated algorithms are solutions to the problem with KT Distance or with top-r Disagreement, or both.
> Is Algorithm 2 used for both Theorem 1 and 2?"
>
> Yes, our proper learning Algorithm 2 is used for both Theorem 1 and Theorem 2. Let us be more specific.
>
> In the KT distance learning, our goal is to control the expected KT distance between the true LSF and our output ranking hypothesis. Algorithm 2 attains this result by first obtaining a collection of $\binom{k}{2}$ linear classifiers (each one associated with the pair $1 \leq i < j \leq k$ and obtained by solving an instance of the Massart halfspace problem) and then by compressing this collection of vectors $v_{ij}$ into a single matrix $W \in \mathbb{R}^{k \times d}$ using a convex program. This algorithm uses $d \log k / \epsilon$ samples to efficiently compute the parameter matrix $W$ that gets $\epsilon$-close to $\Delta_{KT}$ distance. The key lemma that makes this idea work is Lemma 2 in Line 254 which relates the expected KT distance with the angle metric of the two matrices (see also Equation (1) in Line 139). Our Algorithm 2 essentially gives an upper bound on this angle metric.
> When we shift our objective and our goal is to control the top-r disagreement, we can still apply Algorithm 2 which essentially controls the angle metric (see Line 139). The crucial ingredient that is missing is the relation between the loss we have to control, i.e., the expected top-r disagreement and the angle metric of Equation (1) in Line 139. This relation is one of the main technical contributions of this work and is presented in Lemma 1, which essentially says that the expected top-r disagreement is at most $O(kr)$ times this angle metric. Hence, in order to get top-r disagreement of order $\epsilon$, it suffices to apply our Algorithm 2 with $\epsilon’ = O(\epsilon / (kr) )$. This is essentially how the sample complexity of $\widetilde{O}(dkr/\epsilon)$ is obtained.
> We will make this more clear in the first revision of our work.
>
>
> "What is the intuition behind the power 6 in theorem 1 and 2?"
>
> This result comes from an application of an algorithm that learns halfspaces with Massart noise from [ZSA20]. Since we are reducing the problem of learning LSFs to learning binary linear classifiers with noise, any improvement on this bound for binary classification with Massart noise would also improve the corresponding dependence on $1-2 \eta$ in our setting. Intuitively the reason that there is a power on the $(1-2\eta)$ is that all algorithms for learning halfspaces with Massart noise essentially rely on localizing (conditioning) on an area close to the decision boundary of probability inverse proportional to $\mathrm{poly}(1 - 2 \eta)$.

---

### Meta-Review · Area_Chair_NL8Q · 2022-08-26

**Recommendation:** Accept
**Confidence:** Certain

**Metareview:**

The reviewers are unanimous in their strong positive opinion on this paper.  The authors have given the first efficient algorithms for learning noisy linear sorting functions with theoretical guarantees a relevant and useful problems setup for the NeuRIPS community.  The reviewers consider the paper clear and well-presented and thus this is a natural accept.


**Award:**

No

---

### Decision · Program_Chairs · 2022-09-14

Accept